# RAPID Hand: A Robust, Affordable, Perception-Integrated, Dexterous Manipulation Platform for Generalist Robot Autonomy

**Zhaoliang Wan**[1]  **Zetong Bi**[1]  **Zida Zhou**[2]  **Hao Ren**[1]  **Yiming Zeng**[1]  **Yihan Li**[1]

**Lu Qi**[3]     **Xu Yang**[4]     **Ming-Hsuan Yang**[3]     **Hui Cheng**[1*]

[1] Sun Yat-sen University   [2] Orbot Ltd   [3] University of California, Merced   [4] CASIA

*https://rapid-hand.github.io*

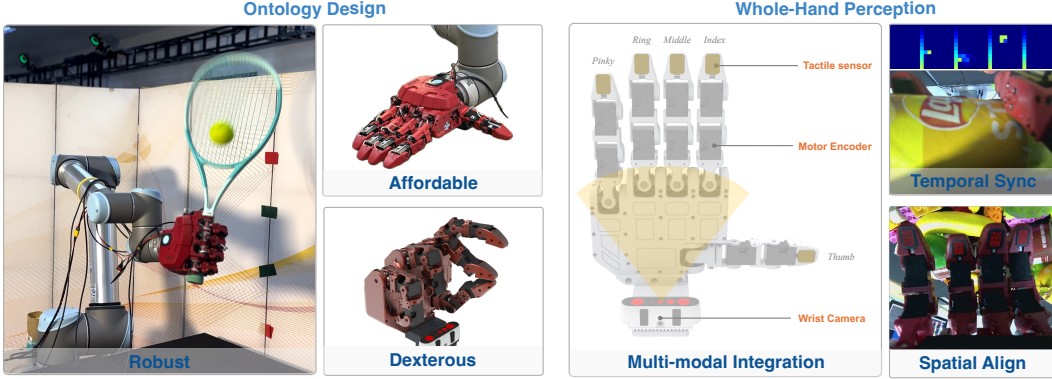

Figure 1: **RAPID Hand** is an open-source, low-cost, fully direct-driven robotic hand platform with stable integrated, synchronized, and aligned multi-modal whole-hand perception.

## Abstract

This paper addresses the scarcity of low-cost but high-dexterity platforms for collecting real-world multi-fingered robot manipulation data towards generalist robot autonomy. To achieve it, we propose the RAPID Hand, a co-optimized hardware and software platform where the compact 20-DoF hand, robust whole-hand perception, and high-DoF teleoperation interface are jointly designed. Specifically, RAPID Hand adopts a compact and practical hand ontology and a hardware-level perception framework that stably integrates wrist-mounted vision, fingertip tactile sensing, and proprioception with sub-7 ms latency and spatial alignment. Collecting high-quality demonstrations on high-DoF hands is challenging, as existing teleoperation methods struggle with precision and stability on complex multi-fingered systems. We address this by co-optimizing hand design, perception integration, and teleoperation interface through a universal actuation scheme, custom perception electronics, and two retargeting constraints. We evaluate the platform's hardware, perception, and teleoperation interface. Training a diffusion policy on collected data shows superior performance over prior works [1, 2], validating the system's capability for reliable, high-quality data collection. The platform is constructed from low-cost and off-the-shelf components and will be made public to ensure reproducibility and ease of adoption.

*Correspondence to: chengh9@mail.sysu.edu.cn

39th Conference on Neural Information Processing Systems (NeurIPS 2025).

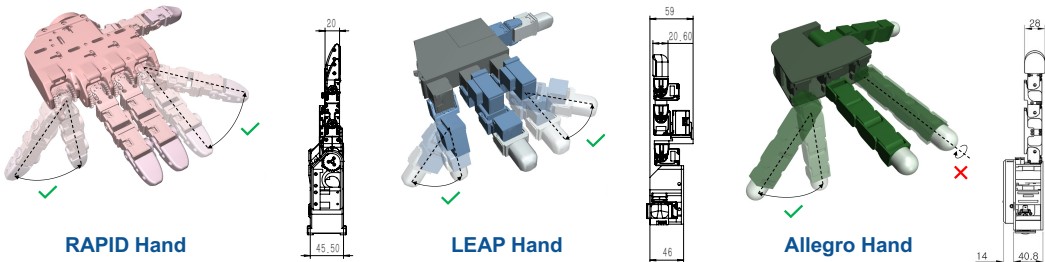

Figure 2: **Finger Size & Kinematics Comparison.** Comparison of RAPID, LEAP, and Allegro in finger thickness, MCP joint configurations, and dexterity, illustrated with representative finger poses.

# 1 Introduction

Dexterous manipulation [3–10] is essential for generalist robot autonomy, benefiting various applications such as household cleaning [11–13] and assistive service [14–17]. Specifically, utilizing large pre-trained vision-language models (VLMs) on high-quality robot-action data [18–24], is emerging as a promising direction for embodied reasoning and general-purpose manipulation. Despite significant progress in model architectures and data curation strategies in such methods, this field still faces two major issues due to limitations in existing hardware platforms used for data collection and policy deployment in general manipulation tasks.

First, due to the limited accessibility of advanced end-effectors, a common practice is to use two-finger parallel grippers, which restricts dexterity in tasks requiring complex fine-grained control, such as in-hand manipulation, tool use [25]. Second, existing multi-finger hardware platforms often focus on mechanical design while overlooking sensor consistency and data integrity. This is observed by a concurrent study [26] that reports a 4.4% dropout rate and a latency of 15-100ms during multi-sensor integration. Both aspects hinder the diversity of manipulation skills and the collection of high-quality real-world demonstration data for generalist robot autonomy.

Collecting high-quality real-world robot demonstrations is challenging due to the lack of a compact and affordable high-DoF hand system for teleoperated manipulation. There are two reasons. On one hand, designing an appropriate actuation and transmission requires elaborate design for motor layout that should balance robustness, low cost, sufficient fingertip force, dexterity, and the risk of bulky structures and unnatural MCP kinematics. On the other hand, finger motions can introduce sensor interference and dropouts, and variable latencies within and across modalities [27]. One question raised: *can we establish a well-structured hand ontology within a seamlessly integrated hardware–software platform for reliable and high-quality collection?*

Then, we build the dexterous manipulation platform from both hardware and software perspectives, ensuring that both components are developed in a consistent manner. For the hardware design, RAPID Hand adopts a compact 20-DoF hand ontology with a universal multi-phalangeal actuation scheme, achieving 20 mm finger thickness through optimized motor layout (Fig. 2). Specifically, this scheme uses direct-drive for distal joints and parallel mechanisms for proximal joints, enabling efficient and independent multi-phalangeal control. Moreover, we propose a hardware-level perception framework to stably integrate wrist-mounted vision, fingertip tactile sensing, and proprioception with precise synchronization (Fig. 5). In software development, we own a high-DoF teleoperation interface, enabling efficient collection of diverse demonstrations across contact-rich tasks. Last, we propose RAPID Hand as a co-optimized hardware–software platform, where the compact 20-DoF hand, robust whole-hand perception integration, and high-DoF teleoperation interface are jointly designed to close the loop from data collection to policy deployment, ensuring durable hardware, stable perception, and efficient, high-quality demonstration collection for dexterous manipulation.

Built upon our dexterous manipulation platform, we validate RAPID Hand by training a conditioned diffusion model on three challenging in-hand manipulation tasks. In our extensive experiments, the method trained on RAPID achieves superior manipulation performance and policy stability (Section 5.3) over prior and concurrent works [1, 2]. To our knowledge, RAPID Hand outperforms existing hands [28, 29] in both ontology design and multimodal perception integration, while remaining low cost and accessible (for more hand comparisons, see Table 2 in Appendix A.1).

The main contributions of this work are as follows:

- **A compact and practical hand ontology for dexterous manipulation.** We design a fully actuated 20-DoF robotic hand with parallel MCP joints and natural human-like kinematics (see Fig. 2). Through extensive prototyping and careful optimization of motor layout and wiring, the finger thickness can be reduced to just 20 mm. This is significantly thinner than previous designs, such as LEAP (59 mm), while improving the robustness of the MCP joint.

- **A hardware-level multimodal integration, synchronization, alignment framework.** To learn contact-rich manipulation, we develop a perception framework that ensures robust multi-sensor integration, precise temporal synchronization, and spatial alignment across both intra- and inter-modalities. This design prevents sensor interference, dropouts, and variable latencies during finger movement, substantially improving the stability and reproducibility of the policy.

- **Learning manipulation skills from whole-hand multimodal perception.** Using a customized high-DoF teleoperation interface, we collect high-quality demonstrations and relax simplified assumptions (e.g., fixed arms, table support) used in prior methods [1] and tackle more challenging tasks such as in-hand translation and rolling, leveraging multiple modalities including vision, touch, and proprioception. In multifinger retrieval, our policy shows significant improvement over concurrent work [2], which relies on single-finger sweeping and ArUco markers for perception.

- **An open and accessible platform for generalist robot autonomy.** RAPID Hand is built from inexpensive, standard, and 3D-printed components in a modular design for easy repair and replacement. We will publicly make all hardware, software, and training pipelines available to support scalable and reproducible research. Unlike closed-source systems like TRX-Hand 5 [27], we prioritize accessibility by balancing affordability and functionality with mass-produced components.

## 2  Related Work

**Dexterous Manipulation for Generalist Autonomy.** Dexterous manipulation is a key capability for enabling generalist robot autonomy. However, current frameworks such as RT-2 [18], GR-2 [7], and $\pi_{0.5}$[24], primarily employ two-finger grippers or low-DoF hands, including GR00T-N1[22] and Gemini Robotics [23], largely due to the high costs and maintenance demands of more advanced end-effectors. This hardware constraint limits manipulation capabilities to gripper tasks, restricting adaptability in fine-grained skills such as in-hand manipulation, tool use, and coordinated multi-finger actions [25]. The scarcity of real-world demonstrations in such complex scenarios further underscores the need for accessible, high-DoF platforms that support contact-rich skill acquisition.

**Hardware Platforms for Multi-fingered Manipulation.** Existing multi-fingered hands, including tendon-driven (e.g., Shadow Hand [30], Faive Hand [31]), direct-driven (e.g., Allegro Hand [29], LEAP Hand [28]), and linkage-driven (e.g., Inspire Hand [32], Ability Hand [33]) designs, face inherent trade-offs among dexterity, robustness, and maintainability. More advanced platforms such as TRX Hand [34, 27] and DLR Hand [35] offer advanced capabilities but remain costly and are for internal use only, limiting broader deployment (see Table 2). Similarly, while various tactile sensing technologies, including piezoresistive [34, 36], optical [37, 38], and magnetic [39] sensors, enable fine-grained contact feedback, they often require stable integration and suffer from wear and tear. These factors constrain the scalability of current platforms for real-world dexterous manipulation, pointing to the need for affordable, robust hands with seamless multimodal sensing integration.

**Learning Dexterous Skills from Teleoperation.** Teleoperation remains a common strategy for collecting robot demonstration data [22], yet applying it to multi-fingered dexterous tasks remains challenging due to the human–robot embodiment gap. Most teleoperation frameworks, whether vision-based [40–42] or vision–touch integrated [25, 15, 43], are built around low-DoF hands with generic retargeting methods [44], limiting their effectiveness in capturing complex, contact-rich behaviors. Tasks such as in-hand translation and rotation are particularly difficult, as these interfaces often result in unstable grasps and object failures. Methods like TILDE [1] attempt to simplify the problem via constrained arm motion and tabletop setups, but these constraints limit applicability to free-space dexterous manipulation. Overcoming these issues requires teleoperation interfaces and platforms that directly address the challenges of high-DoF retargeting, perception integration, and stable data collection in complex manipulation scenarios.

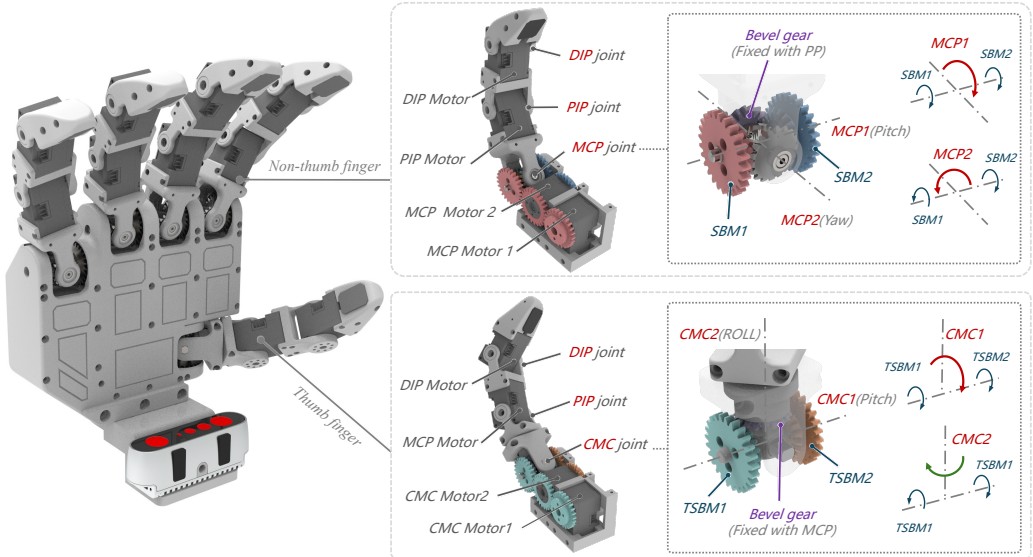

Figure 3: **RAPID Hand with universal multi-phalangeal actuation scheme.** The DIP and PIP joints of the non-thumb fingers, and the DIP and MCP joints of the thumb, are directly driven by segment-mounted motors. The MCP joints of the non-thumb fingers and the CMC joint of the thumb are actuated via parallel mechanisms, enabling independent control.

## 3 RAPID Hand Platform

We present RAPID Hand, a cost-effective humanoid hand designed for generalist robot autonomy. The design focuses on two aspects: *hand ontology*, balancing robustness, affordability, fingertip force, and dexterity; and a *whole-hand perception framework*, integrating multimodal sensors with spatial alignment and temporal synchronization. Additional hand evaluation and analysis details can be found in the supplementary material.

### 3.1 RAPID Hand Ontology

**Hand Kinematics.** The ontology design emphasizes anthropomorphic dexterity, enabling seamless interaction with household objects and tools crafted for human use. To achieve this, we require ours to imitate the natural appearance and intricate kinematics of the human hand that usually has 20 to 22 DoFs. The human hand has five fingers, each with specific joint structures [45]. The thumb includes interphalangeal (IP), metacarpophalangeal (MCP), and trapezoid-metacarpal (TM) joints, while the other fingers contain distal (DIP), proximal (PIP), and MCP joints. To replicate the ball-and-socket structure of the MCP and TM joints, we replace each with two hinge joints, with the remaining joints simplified to single-axis hinge types (Fig. 15). The design of the RAPID Hand incorporates 20 motors, four for each finger to allow for precise control over complex movements. A kinematic comparison between the RAPID Hand, LEAP Hand, and Allegro Hand is presented in Fig. 2.

**Hand Ontology Design.** In addition to human-like dexterity, the design of our finger ontology emphasizes robustness, affordability, and sufficient fingertip force. To meet these requirements, we introduce a **universal multi-phalangeal actuation scheme** for the fingers. Cost-effectiveness is a key consideration, which necessitates using commercially available servo motors instead of custom brushless motors. However, this choice presents challenges due to the larger size of these motors. Building on the foundation of cost-effectiveness, the multi-phalangeal structure aims to achieve dexterity by allowing four DoFs per finger. Achieving this while maintaining anthropomorphic kinematics and appearance is challenging when using off-the-shelf servo motors to drive the joints. For instance, the MCP joint, which functions as a ball joint, can be approximated with two DoFs: abduction/adduction and flexion/extension. While some designs (e.g., [29, 28]) use separate motors positioned in various orientations to achieve these DoFs, they often compromise the natural kinematics and appearance of the hand. Our earlier prototype [46] explores a fully actuated high-DoF design with all actuators placed in the palm, which makes the palm bulky and weakens fingertip force.

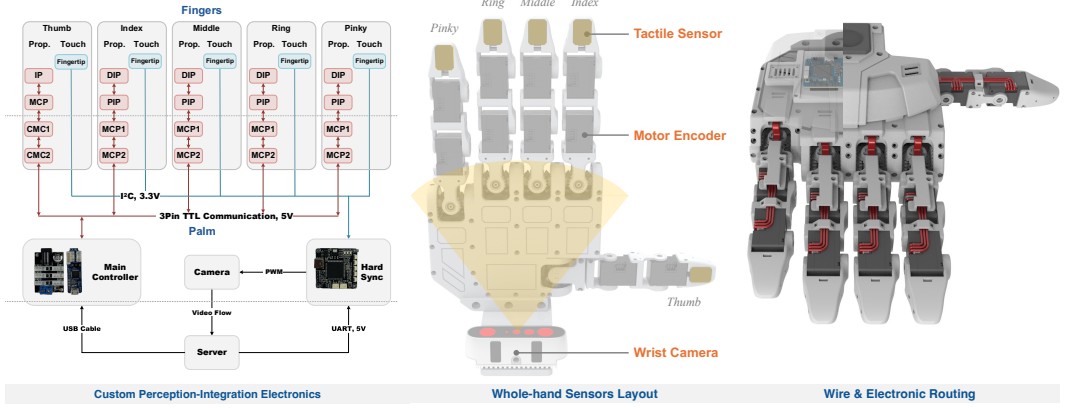

Figure 4: **Whole-hand perception framework** including vision, tactile, and proprioceptive sensor layout with electronics and wiring routing.

To tackle this challenge, we implement a bevel gear differential mechanism in the MCP joint, using off-the-shelf servo motors. We design a spur-bevel gear module (SBM), which integrates a spur gear and a bevel gear, replacing a traditional belt-driven system with a gear-based system. As illustrated in Fig. 3, two motors (MCP-1, MCP-2) drive SBM-1 and SBM-2 through parallel shaft gear sets. These two modules engage with the bevel gear fixed on the proximal phalanx (PP), allowing the rotation of the motors to dictate the movement of the MCP joint. The motion control in the MCP joint can be broken down into three categories:

- `Flexion/Extension`: When SBM-1 and SBM-2 rotate at the same speed and direction, the MCP joint rotates along the MCP-1 axis (pitch), causing the finger to flex or extend.

- `Abduction/Adduction`: When SBM-1 and SBM-2 rotate at the same speed but in opposite directions, the MCP joint rotates along the MCP-2 axis (yaw), enabling the finger to move side-to-side.

- `Combined Motion`: By coordinating the rotations of SBM-1 and SBM-2 that control both axes of the joint, the MCP joint can simultaneously flex, extend, and move side-to-side.

The differential mechanism allows for the control of a single joint with multiple DoFs, effectively simulating the flexibility of the MCP joints in the human hand. It also enhances the robustness of the MCP joint (see further details in Appendix A.1.1), minimizes the space required compared to belt-driven systems, and eliminates issues related to belt-tension, resulting in a more compact and reliable finger mechanism. To ensure adequate fingertip force and robust performance, we embed the same motors for the PIP and DIP joints within the proximal and intermediate phalanges, respectively. This configuration enables each joint to operate independently, improving both force output and the overall durability of the system.

This design is compatible with non-thumb fingers and requires only minor adjustments for the thumb. A differential mechanism is employed in the CMC joint for the thumb finger. Two thumb spur-bevel gear modules (TSBM) mesh with a bevel gear that is fixed to the thumb finger, allowing the CMC joint to rotate along the CMC-1 axis (pitch) and the CMC-2 axis (roll). This arrangement of DoFs effectively simulates the movement of the CMC joint in the human palm.

## 3.2    Whole-Hand Perception Framework

**Multi-modal Perception Integration.** Integrating stable whole-hand perception into high-DoF hands presents practical challenges, particularly as finger movements can cause wiring interference and sensor dropouts. Existing solutions, such as the hands described in [27, 34, 47], provide whole-hand tactile coverage but suffer from high complexity, closed-source design, and prohibitive costs. Optical tactile sensors [43] offer higher resolution at lower cost but are prone to uniformity issues and degradation of the gel layer over time. Similarly, systems like the Ability Hand [25] integrate wrist vision and fingertip FSR sensors, yet the tactile feedback lacks the sensitivity and consistency

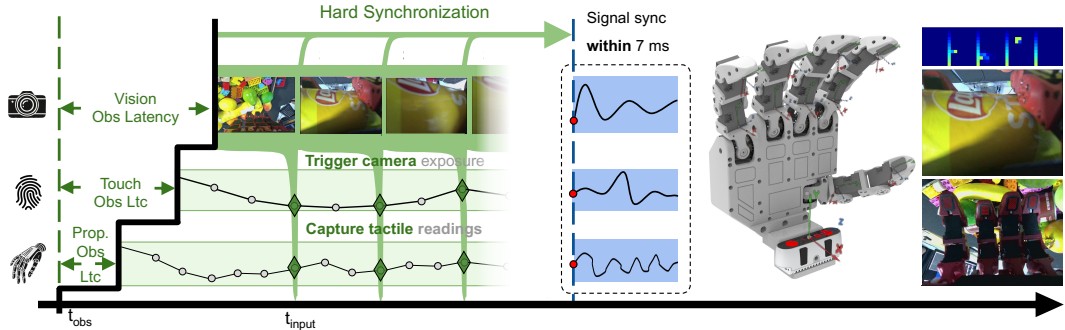

**(a) Multi-modal Integration**     **(b) Temporal Synchronization**     **(c) Spatial Alignment**

Figure 5: **Whole-hand Multimodal Perception Alignment.** (a) RAPID Hand observation collects a sequence of **unsynchronized** data, including RGB images, tactile signals, and 20 joint angles with inconsistent latencies. (b) We temporally synchronize the various observation streams using a hard-sync framework. (c) The results are a sequence of temporally synchronized and spatially aligned whole-hand multimodal perceptions.

required for reliable manipulation data collection. These limitations highlight the need for a more practical, scalable approach to perception integration.

To address these challenges, the RAPID Hand adopts an economical yet robust scheme combining wrist-mounted vision and fingertip tactile sensing (Fig. 4 center). Specifically, we integrate an RGBD camera (Orbbec[48]) and flat piezoresistive tactile sensors (Matrix [49]) at the fingertips, providing global in-hand observation and localized contact feedback. The tactile sensors, offering 96 taxels per fingertip, achieve sufficient sensitivity, resolution, and long-term durability for dexterous manipulation tasks, while remaining affordable and accessible for the broader community. To ensure stable and synchronized multi-modal data, we develop custom electronics (Fig. 4 left), supporting reliable integration of vision, touch, and proprioception streams. Wire and electronics routing are carefully optimized (Fig. 4 right) to minimize protrusions and avoid restricting finger motion, while improving modularity, ease of maintenance, and system robustness during extended operation. Through these design decisions, RAPID Hand achieves a balance between affordability, durability, and whole-hand perception capability, providing a scalable alternative to costly, closed-source systems and supporting robust dexterous manipulation research.

**Hardware Temporal Synchronization.** Beyond integrating multiple sensors into the hand, synchronizing their signals temporally is essential for embodied manipulation tasks, particularly when training policies from demonstration data. Inconsistent latencies across or within modalities can degrade data quality, necessitating precise synchronization to ensure reliability. Previous research [13] tackles this issue with software-based synchronization (soft-syncing vision and proprioception), which performs well for in-the-wild data collection. However, this approach may be less robust in teleoperation or sim-to-real settings. In contrast, hardware solutions like those described in [34] achieve precise spatial alignment for in-hand perception but depend on a stationary setup for temporal synchronization, which limits their effectiveness in dynamic manipulation scenarios.

Our system addresses these challenges through a **hard-sync framework** integrated into our custom perception-integration electronics system. Control commands for the 20 motors are dispatched through the main controller, while a dedicated sync module is simultaneously:

- Captures fingertip tactile readings via $I^2C$ (within $\leq$ 7 ms).
- Triggers wrist camera exposure via PWM (within $\leq$ 2 ms).

This tightly coordinated approach ensures that multimodal data streams are aligned within $\leq$ 7 ms, facilitating reliable real-time data collection (Fig. 5 (b)) at a rate of 25 Hz—a key requirement for dynamic in-hand manipulation tasks.

**In-Hand Spatial Alignment.** The spatial alignment of multimodal sensor data (Fig. 5 (c)) is essential to complement temporal synchronization, ensuring coherent perception for in-hand manipulation.

- `Proprioception`: The joint angles for all 20 DoFs are calibrated to sub-degree precision, thanks to a robust mechanical design (refer to the joint accuracy analysis in 3.1).

- `Touch:` Fingertip tactile arrays capture local 3D contact geometry, which is critical when external cameras are occluded. Forward kinematics transforms these signals into a "local touch point cloud" using calibrated joint states and taxel positions.
- `Vision:` A wrist-mounted camera provides global object context. Tactile and visual data are aligned to the vision coordinates by calibrating the camera's intrinsic and extrinsic and aligning taxel positions to the hand's kinematic model.

Unlike previous research on tactile-based manipulation that relies solely on raw tactile readings [47], our approach generates spatially aligned touch point clouds that provide direct geometric correspondence. As shown in Fig. 1, all fingertip taxels map precisely to the hand's frame, resulting in synergy between vision, touch, and proprioception, similar to human manipulation. This integrated perception framework allows for reliable whole-hand perception of objects, even under occlusion.

## 4 Learning Dexterous Skills

To validate the RAPID Hand platform and the collected data, we train a whole-hand visuotactile policy on three challenging in-hand manipulation tasks: object-in-hand translation, rolling, and multi-fingered nonprehensile retrieval.

### 4.1 High-DoF Teleoperation Interface

We first develop a high-DoF teleoperation interface (Fig. 20), where human hand poses are captured using an Apple Vision Pro and retargeted in real time to a UR10e arm equipped with RAPID Hand. During demonstrations, we record synchronized in-hand RGB images, tactile readings, joint angles, and taxel spatial positions via RAPID Hand's perception framework.

A key challenge in teleoperating high-DoF end-effectors is bridging the embodiment gap between human and robotic hands. Existing methods commonly approximate the robot hand by uniformly scaling the human hand skeleton, but this suffers from two critical limitations. First, uniform scaling introduces geometric distortions, as it fails to account for mismatched link lengths and joint limits between the human and robot hands. Second, by neglecting inter-finger coupling, these methods often produce functionally inconsistent motions, where essential synergies, such as thumb–finger pinch closures, are poorly replicated.

To address these challenges, we introduce two additional constraints into the retargeting optimization: a *conformal-aligned constraint* that enforces local geometric alignment, and a *contact-aware coupling constraint* that adaptively reinforces inter-finger coordination during critical interactions. Specifically, we apply per-phalangeal segment geometric calibration to align each human keypoint with its corresponding robot link, thereby reducing local kinematic discrepancies and mitigating global-scale distortions. To improve contact consistency during manipulation, we further introduce an interaction-aware thumb–finger coupling term, whose influence increases smoothly as the fingertips approach each other. Finally, a lightweight temporal smoothing prior is incorporated to suppress residual jitter while preserving system responsiveness. Together, these constraints enable spatially accurate, contact-consistent, and temporally stable retargeting without requiring manual parameter tuning:

$$\min_{q(t)} \lambda_1 \underbrace{\sum_{(i,j)\in\mathcal{K}}\left\|v_{i,j}(t)-\mathrm{FK}_{i,j}(q(t))\right\|^2}_{\text{Conformal-aligned Constraint}}+\lambda_2\underbrace{\sum_{i\in\mathcal{I}}\omega_i(t)\left\|\Delta_i(t)-g_i(q(t))\right\|^2}_{\text{Contact-aware Coupling Constraint}}+\lambda_3\underbrace{\left\|q(t)-q(t-1)\right\|^2}_{\text{Temporal Smoothness}}, \quad (1)$$

where $q(t) \in \mathbb{R}^n$ represents the joint angle vector of the RAPID Hand at time $t$. See Appendix A.2.1 for detailed method description and symbol explanation.

### 4.2 Whole-Hand Visuotactile Policy

Based on the collected demonstrations, we train a whole-hand visuotactile policy using a diffusion-based generative model [50] (Fig. 19). The policy takes as input wrist images, fingertip tactile readings, and proprioception, all temporally synchronized and spatially aligned through our perception

framework (Fig. 5). Taxel readings are embedded together with their spatial positions; vision and proprioception are processed via pre-trained encoders and MLPs. These modalities are fused into a unified representation, enabling the policy to predict future 26-DoF joint trajectories for both hand and arm. We train the policy for 300 epochs on collected data and deploy it at 10 Hz.

## 5 Experimental Results

We comprehensively evaluate the RAPID Hand platform to validate its hardware robustness, whole-hand perception consistency, and effectiveness in supporting dexterous manipulation learning. We further conduct ablation studies to analyze the impact of perception integration on policy performance and robustness under sensor dropouts and latency.

### 5.1 Hardware Platform Evaluation

**Proprioceptive and Tactile Performance.** We first assess the proprioceptive performance of the RAPID Hand under both unloaded and loaded conditions. Sinusoidal tracking tests on the index and thumb joints confirm stable position tracking without performance degradation or overheating during continuous operation (Fig. 8). Under load, extending beyond LEAP's MCP joint error test with a 25g weight [28], we apply 100g and 200g weights to the middle finger and measure MCP joint positional errors. The results suggest the system maintains acceptable precision even under external forces (Fig. 9). Additionally, the parallel MCP joint design improves load tolerance compared to the LEAP Hand, potentially enhancing safety and robustness in contact-rich tasks. These benefits can be attributed to the universal multi-phalangeal actuation scheme and the optimized motor arrangement (Fig. 11).

For tactile sensing, we evaluate sensitivity and consistency using a three-axis calibration stage (Fig. 10). Sensitivity tests indicate the sensors can detect forces as low as 0.39g (bare sensor) and 0.59g (with protective cover). For consistency, a 3kg distributed load across 15 sensors shows deviations between -4% and 8%, suggesting sufficient uniformity for reliable manipulation.

**Hand Specifications and Dexterity Comparison.** The RAPID Hand delivers up to 7N fingertip force, which is sufficient for most dexterous manipulation tasks [27]. Its overall size is comparable to LEAP and Allegro Hands, with the exception of the additional pinky finger (Fig. 13). The system weighs 1.148 kg, slightly heavier than Allegro (1.08 kg) and LEAP (0.748 kg) due to the added pinky and integrated sensors. Similar to LEAP and Allegro, most joints are directly driven, while the MCP joints adopt a gear mechanism with an efficiency of approximately 96–98%.

We evaluate quantitative dexterity using standard metrics, including thumb opposability (Table 3, Fig. 14) and manipulability (Table 4). Given the similar hand sizes, these metrics allow a fair comparison. The RAPID Hand achieves a balanced trade-off between human-like kinematics and dexterity, supporting more natural retargeting and hand poses (Fig. 16).

For qualitative dexterity, RAPID Hand successfully replicates all 33 grasp types in the Feix taxonomy [51], demonstrating its versatility in power, intermediate, and precision grasps (Fig. 22). We further compare in-hand translation tasks using the teleoperation method from [44]. As shown in Fig. 17, RAPID Hand enables more natural lateral finger motions, while Allegro easily drops objects, and LEAP shows minimal motion. These results illustrate RAPID Hand's capability to support human-like grasping and manipulation behaviors, facilitating more efficient collection of robot demonstrations.

### 5.2 Software Interface Evaluation

**Qualitative Evaluation of Retargeting Optimization.** We visualize the retargeting results in Fig. 18 and Fig. 23 to qualitatively assess retargeting constraints. In Fig. 18, we visualize the retargeting process step by step. Column (a) shows the human hand performing the target gestures. Colunm (b) presents the corresponding 3D keypoints and pose estimated by MediaPipe [52]. In column (c), applying our conformal alignment constraint corrects these distortions, producing geometrically consistent human hand poses. Column (d) illustrates the resulting retargeted robot poses optimized by our method, while column (e) shows the RAPID Hand executing these poses, closely matching the intended geometry without visible distortions. Fig. 23 further compares our method with a uniform

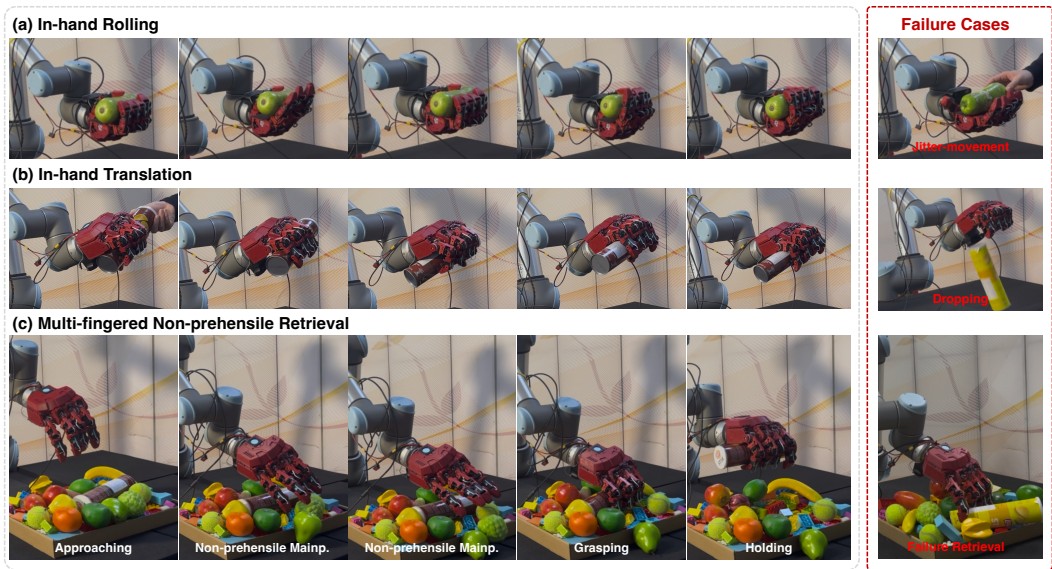

Figure 6: Dexterous Manipulation Performance.

scaling baseline across four scale factors $\alpha$. None of the $\alpha$ values achieves consistent alignment across all gestures: smaller scales under-extend longer fingers, while larger scales over-extend distal links and still fail to close thumb–finger gaps. In contrast, our method, using a single parameter set, accurately reconstructs all poses with correct link lengths and reliable thumb–finger contact, demonstrating the effectiveness of our local geometric alignment and contact-aware coupling.

## 5.3 Dexterous Skills Evaluation

**Policy Learning Performance.** Using the high-DoF teleoperation interface, we collect demonstrations and train a whole-hand visuotactile policy on three challenging tasks: translation, rolling, and non-prehensile retrieval. The policy achieves near-perfect success rates on rolling and translation tasks, while retrieval remains challenging due to longer sequences and complex interactions (Table 1). Notably, compared to prior and concurrent works, our platform relaxes common assumptions such as fixed arms or table support [1], enabling more generalizable in-hand rolling and translation. For retrieval, our multifinger policy substantially outperforms concurrent methods [2], which rely on single-finger sweeping and ArUco-based perception. Ablation studies further highlight the complementary roles of vision, touch, and proprioception across tasks. Removing any modality notably degrades performance, particularly in scenarios requiring fine contact adjustments.

**Robustness under Delays and Dropouts.** We further introduce controlled perception delays and random dropouts during policy execution. As shown in Fig. 7 and Table 1, stable perception integration and synchronization significantly enhance policy stability and reduce action errors, confirming the effectiveness of the RAPID Hand platform in ensuring robust policy deployment.

**Generalization to Unseen Objects.** We test the learned policy on unseen objects such as corn, wine bottles, and chip bags. The policy generalizes well without task-specific retraining, demonstrating the benefits of reliable whole-hand perception and consistent data collection pipelines (Fig. 21).

| Task | Rolling | Translation | Retrieval |
|---|---|---|---|
| w.o. Vision | 3 / 50 | 8 / 50 | 0 / 50 |
| w.o. Touch | 6 / 50 | 9 / 50 | 22 / 50 |
| w.o. Prop. | 4 / 50 | 5 / 50 | 28 / 50 |
| whole-hand policy | 50 / 50 | 50 / 50 | 24 / 50 |
| w. 4.4% dropout | 44 / 50 | 41 / 50 | 23 / 50 |

Table 1: Success Rate on the Three Manipulation Tasks

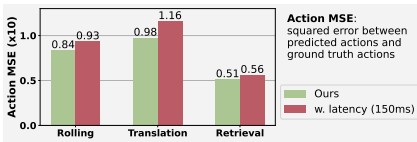

Figure 7: Action MSE w./w.o. latency.

# 6 Conclusion

This work presents RAPID Hand, an affordable and perception-integrated 20-DoF robotic hand designed to facilitate high-quality data collection for dexterous manipulation. Through co-optimization of hand ontology, perception integration, and teleoperation interface, we address key challenges in controlling high-DoF hands and collecting reliable multi-modal data. Experiments on in-hand manipulation tasks demonstrate the platform's effectiveness in supporting policy learning, with improved performance and stability over prior systems. While promising, the current design remains constrained by servo motor size and lacks direct haptic feedback. Future work will explore more compact actuators and enhanced teleoperation interfaces to further improve usability and precision.

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

Table 2: **Comparison of Multi-Fingered Hands.** An overview of widely used multi-fingered hands in recent dexterous manipulation research, detailing their finger count, in-hand perception capabilities, actuation types, and other specifications, including cost, DoFs, and alignment and synchronization availability. The symbols ✓ and - indicate the presence or absence of a modality or feature, respectively, while ○ denotes unclear.

| Hands | Finger Num. | In-hand Perception | | | Types of Actuation | | | Other Specifications | | |
|---|---|---|---|---|---|---|---|---|---|---|
| | | Vision | Touch | Prop. | Tendon | Direct | linkage | Cost (USD) | DoFs | Align or Sync |
| Barrett Hand [53] | 3 | - | ✓ | ✓ | - | ✓ | - | 50,000 | 4 | - |
| TRX Hand [34] | 3 | - | ✓ | ✓ | ✓ | - | - | ○ | 8 | ✓ |
| EyeSight Hand [43] | 3 | ✓ | ✓ | ✓ | - | ✓ | - | 2,500 | 7 | ✓ |
| Allegro Hand [29] | 4 | - | - | ✓ | - | ✓ | - | 16,000 | 16 | - |
| LEAP Hand [28] | 4 | - | - | ✓ | - | ✓ | - | 2,000 | 16 | - |
| DLR Hand II [35] | 4 | - | ✓ | ✓ | - | ✓ | - | ○ | 12 | - |
| Delta Hand [1] | 4 | ✓ | - | ✓ | - | - | ✓ | 1,000 | 12 | ✓ |
| Shadow Hand [30] | 5 | - | ✓ | ✓ | ✓ | - | - | 300,000 | 20 | ○ |
| TRX Hand 5 [27] | 5 | - | ✓ | ✓ | ✓ | - | - | ○ | 13 | - |
| Faive Hand [31] | 5 | - | - | ✓ | ✓ | - | - | ○ | 11 | - |
| Inspire Hand [32] | 5 | - | - | ✓ | - | - | ✓ | 5,000 | 6 | - |
| Ability Hand [33] | 5 | - | ✓ | ✓ | - | - | ✓ | 20,000 | 6 | - |
| RAPID Hand | 5 | ✓ | ✓ | ✓ | - | ✓ | - | 3,500 | 20 | ✓ |

# A  Appendix

## A.1  Hand Analysis

### A.1.1  Robustness

**Reliability**. The RAPID Hand employs current-based position control for precise joint actuation. As shown in Fig. 8, sinusoidal input tests for both the index finger and thumb joints demonstrate stable positional tracking with minimal deviation. Throughout repeated trials, the hand consistently maintains its accuracy without performance degradation or overheating. This reliability is attributed to the universal multi-phalangeal actuation scheme and the optimized arrangement of the motor, wiring, and electronics. These features facilitate extensive real-world data collection and effective policy deployment, demonstrating the robustness of the ontology design.

**Fingertip Force**. We measured fingertip force by having the RAPID Hand apply pressure with its index finger onto a 6D force sensor, which recorded measurements of up to 7 N for the index finger. Given that most daily items weigh less than 1 kg [54], this force is sufficient for most tasks requiring dexterous manipulation. Additionally, pull-push tests demonstrate that the parallel MCP joint design supports a load capacity 2.3 times greater than that of the LEAP Hand's tandem configuration, ensuring robustness during forceful interactions.

### A.1.2  Affordability

**Fabrication Costs.**

The RAPID Hand offers a cost-effective platform for researchers in dexterous manipulation. Its primary components—off-the-shelf motors and sensors, 3D-printed parts, a main controller, and other electronics—total approximately $3,500. The majority of this expense comes from the 20 DYNAMIXEL servo motors, which are commonly used in previous studies [28, 31, 55, 56]. For researchers who already possess LEAP Hands, upgrades primarily require just four additional motors ($360), significantly lowering costs. This cost efficiency, combined with the system's modularity, makes the RAPID Hand an accessible platform for scalable research in dexterous manipulation.

**Maintenance.** The RAPID Hand features an open-source design optimized for self-maintenance, which addresses the challenges of long-term use, such as inevitable wear and tear on motors and sensors. Its modular multi-phalangeal architecture allows researchers to easily replace or repair components without relying on factory services, thereby reducing downtime and costs—common limitations of commercial robotic hands. This capability for self-maintenance is particularly valuable

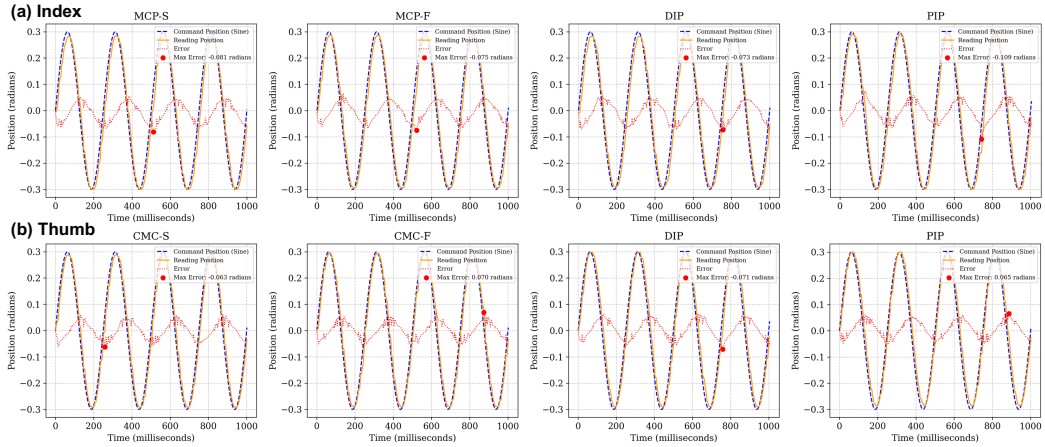

Figure 8: **Accuracy of the index and thumb finger.**

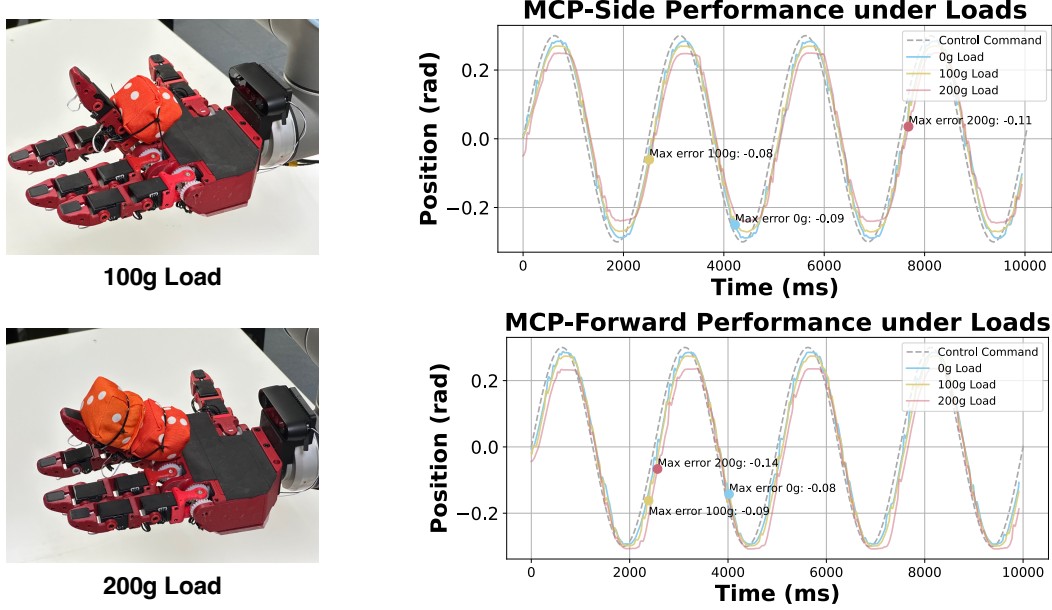

Figure 9: **Accuracy of the middle finger under loads.**

for continuous data collection and policy deployment in embodied manipulation, where interruptions can significantly impede progress. Compared to tendon-driven or linkage-driven hands, the RAPID Hand's design simplifies the repair process, empowering researchers to maintain their hardware efficiently and focus on advancing their algorithms.

### A.1.3    Perception-Integration.

**Whole-hand Perception.** The RAPID Hand provides researchers with a practical and cost-effective alternative for whole-hand perception. While full-coverage tactile sensors are ideal, they remain prohibitively expensive due to the complexity of integrating multi-curved surfaces with robust tactile performance, as shown in Fig. 12. To accommodate printed circuits, large surfaces must often be partitioned into smaller, developable ones, further increasing cost and design complexity. To address this, the RAPID Hand integrates wrist-mounted vision with mass-produced flat tactile sensors on its fingertips, offering a scalable and robust solution for whole-hand perception. Especially since the

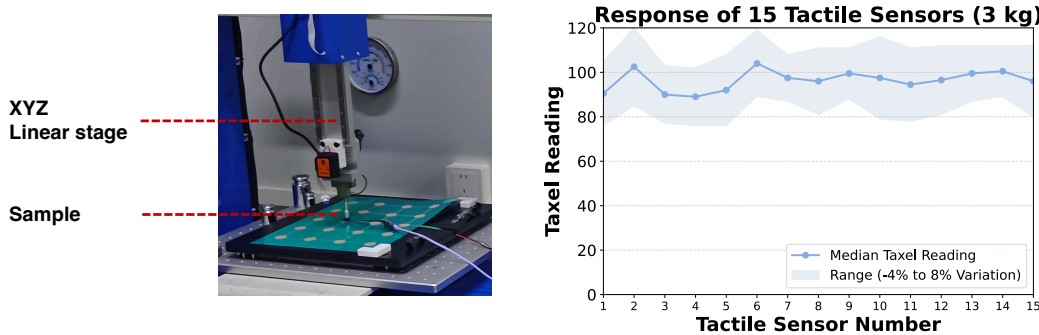

Figure 10: **Tactile performance evaluation.**

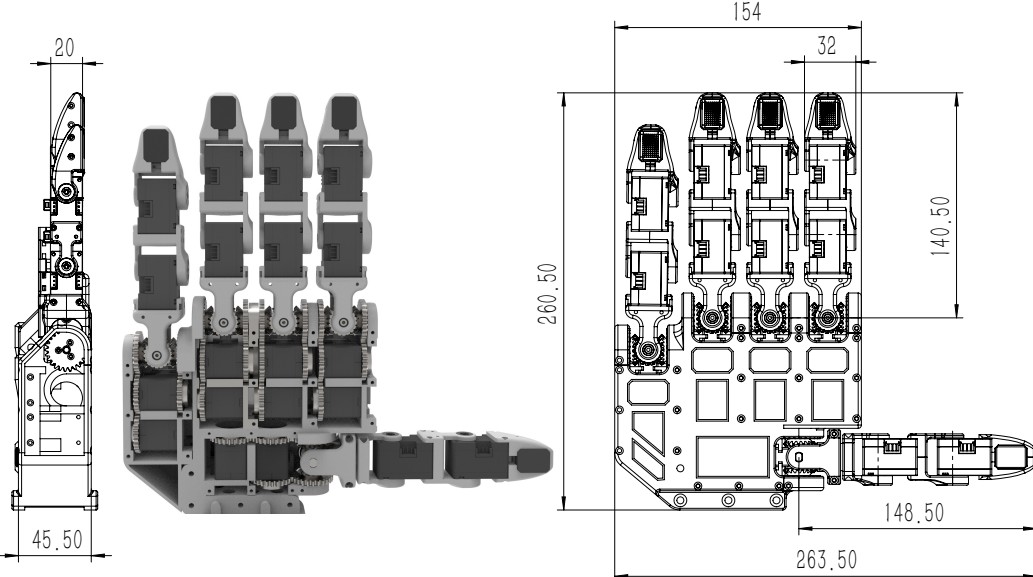

Figure 11: **Optimized Motor Arrangement Design.** The finger thickness is reduced to 20 mm, significantly thinner than LEAP's 59 mm.

tactile sensors are inevitable to wear and tear after long-term use, the five tactile sensors ($500 in total) are affordable.

**Whole-hand Perception Alignment.** The RAPID Hand aligns vision, touch, and proprioception both temporally and spatially, supporting the requirements for large-scale manipulation data collection. Temporal synchronization ensures consistent and stable multi-sensor data streams, while precise calibration maintains spatial alignment, allowing for accurate whole-hand perception.

Reliable multi-modal alignment is particularly important for touch sensing, where fine-grained spatial accuracy is needed to capture local 3D contact information. While the TRX Hand [34] achieves spatial alignment by collecting stationary in-hand data, the RAPID Hand is designed for real-time synchronization and alignment in dynamic tasks, making it a potentially useful tool for embodied manipulation research. The system achieves hardware-level synchronization within a 7 ms latency and pixel-level spatial accuracy, ensuring consistency in perception data, helping to improve data quality, and preventing sensor dropouts occasionally or latency inconsistencies during collection.

This whole-hand multimodal perception system could also contribute to reinforcement learning (RL)-based manipulation methods [37, 57, 58], where policy training benefits from well-aligned multi-modal data. In Sim2Real transfer, precise spatiotemporal alignment is particularly crucial for tactile-based manipulation, where discrepancies between simulated and real-world feedback remain a significant challenge. To help address this, the RAPID Hand includes a tactile simulation environment

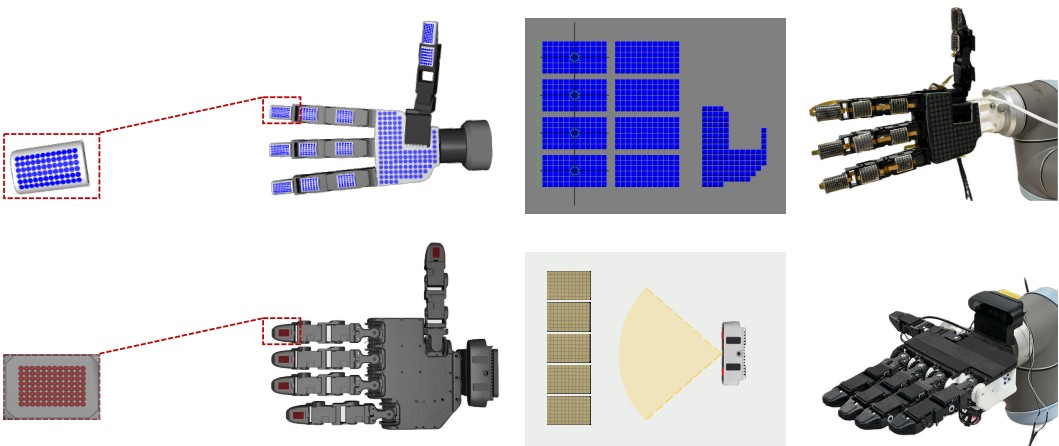

Figure 12: **Simulated and real-world hands with whole-hand perception**. The figure compares tactile sensing in the four-fingered Allegro (top) and the five-fingered RAPID (bottom). The Allegro Hand integrates full-coverage tactile sensors, which are costly and may degrade over time. In contrast, the RAPID Hand offers a cost-effective alternative, combining fingertip tactile sensors with a wrist-mounted camera for whole-hand perception. Red points indicate 480 simulated taxels in MuJoCo, with activated sensor signals and positions recorded as tactile readings.

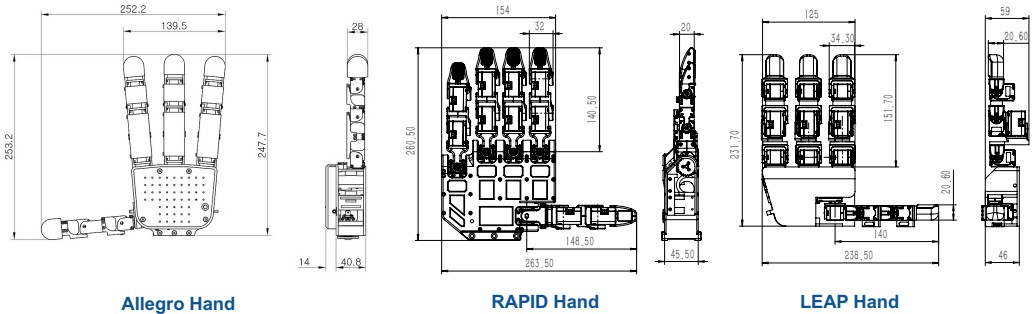

**Allegro Hand**          **RAPID Hand**          **LEAP Hand**

Figure 13: **Visualization of hand sizes.**

Table 3: **Finger-to-Thumb Opposability Volume** (mm$^3$)

| Hands | Index | Middle | Ring | Pinky |
|---|---|---|---|---|
| Allegro Hand | 320,388 | 265,961 | 107,904 | - |
| LEAP Hand | 989,013 | 829,705 | 534,242 | - |
| RAPID Hand (Ours) | 312,233 | 317,805 | 252,510 | 144,970 |

based on MuJoCo [59], as shown in Fig. 12. This simulation focuses on accurate contact position feedback rather than replicating the non-linear force responses of individual taxels. Each taxel is modeled as a force sensor, positioned based on real-world 3D scans. Upon contact, the system records the activated sensor positions as tactile reading signals, using the hand's kinematics, and estimates their relative poses within the hand's reference frame. While challenges remain, the RAPID Hand's design aims to help bridge the gap between simulated and real-world tactile interactions, contributing to further research in this area.

### A.1.4 Dexterity

**Quantitative Dexterity.** To fairly compare the dexterity of the Allegro Hand, LEAP Hand, and RAPID Hand, we adopt the evaluation metrics used in [29, 28]: thumb opposability and manipulability. **Thumb opposability**[60], a key factor for in-hand manipulation, is summarized in Table 3 and visualized in Fig. 14. The LEAP Hand exhibits the highest opposability due to its MCP joint

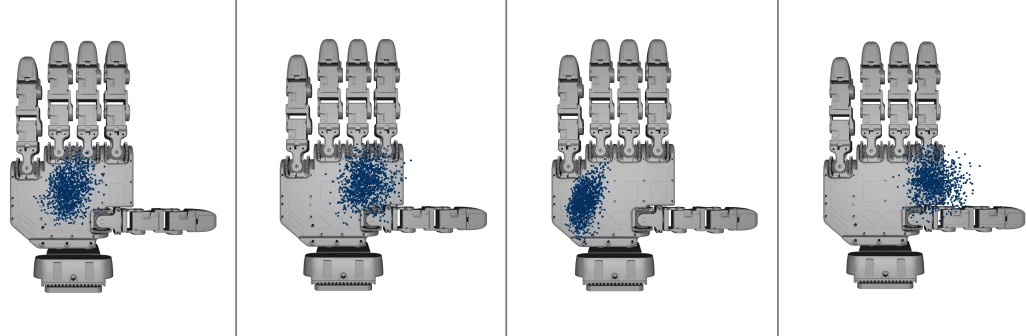

Figure 14: **Visualization of Thumb Opposability Volume in the RAPID Hand.** Blue points indicate the possible positions within the opposability range from each non-thumb finger to the thumb of the RAPID Hand.

Table 4: **Manipulability Ellipsoid Volume** (mm$^3$)

| Hand/Finger Pose | Down | Up | Curled |
|---|---|---|---|
| *Allegro Hand* | | | |
| Linear | 246 | 48.3 | $2.21 \times 10^5$ |
| Angular | 0 | 0 | 0 |
| *LEAP Hand* | | | |
| Linear | $3.03 \times 10^3$ | $3.03 \times 10^3$ | $1.36 \times 10^5$ |
| Angular | $1.18 \times 10^3$ | $5.23 \times 10^5$ | $2.50 \times 10^5$ |
| *RAPID Hand (Ours)* | | | |
| Linear | $3.01 \times 10^4$ | $3.03 \times 10^4$ | $1.77 \times 10^5$ |
| Angular | $3.24 \times 10^4$ | 20.8 | $2.84 \times 10^4$ |

placement on the second phalangeal segment. However, this design results in a bulkier appearance with unnatural kinematics. The RAPID Hand, in contrast, demonstrates significant improvement over the Allegro Hand by optimizing finger design and motor arrangement, achieving a better balance between dexterity and form factor. **Manipulability** [61] measures the hand's dexterity in specific poses. The manipulability ellipsoid volumes for three standard poses—**down, up, and curled**—are listed in Table 4. The RAPID Hand demonstrates improved manipulability in most tested poses compared to the LEAP and Allegro Hands, reflecting its refined ontology design for dexterous tasks.

**Qualitative Dexterity.** The RAPID Hand provides more natural finger poses than the LEAP Hand when retargeting human hand movements to robotic hands. As shown in Fig. 16, direct mapping of human motions to the LEAP Hand often leads to collisions and unnatural finger configurations, particularly during fist closure. In contrast, the RAPID Hand achieves a more anthropomorphic fist pose, improving usability in teleoperation and imitation learning. To further assess its anthropomorphic capabilities, we applied the **Feix taxonomy** [51], which classifies human hand grasp types based on functionality. The RAPID Hand successfully replicates all **33 grasp types** defined in the taxonomy—including power, intermediate, and precision grasps—as illustrated in Fig. 22. These results highlight the RAPID Hand's ability to mimic human grasping strategies, making it well-suited for human-to-robot motion retargeting applications.

### A.1.5  Compatibility

**Compatible Ontology Design**. The RAPID Hand's ontology design follows a universal multi-phalangeal actuation scheme with an optimized motor arrangement, ensuring both anthropomorphic dexterity and compatibility with various motor types. In the RAPID hand, 20 DYMANXIEL servo motors are used as a practical compromise. Although brushless motors offer greater power, their larger size, and higher cost currently limit their use in directly driven 20-DoF hands. With minimal adjustments, the design can accommodate brushless motors, making it adaptable for future

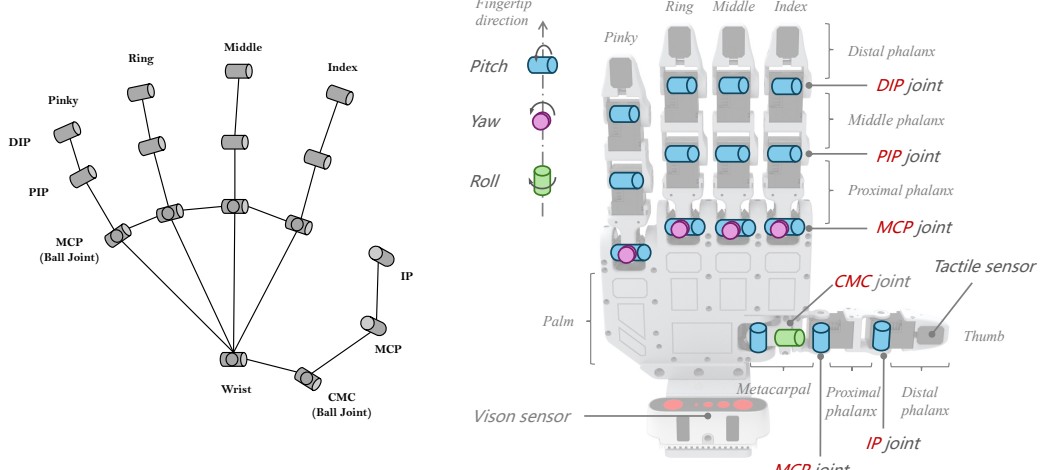

Figure 15: **Hand Kinematics.** Simplified human hand kinematics (left) and RAPID Hand kinematics (right).

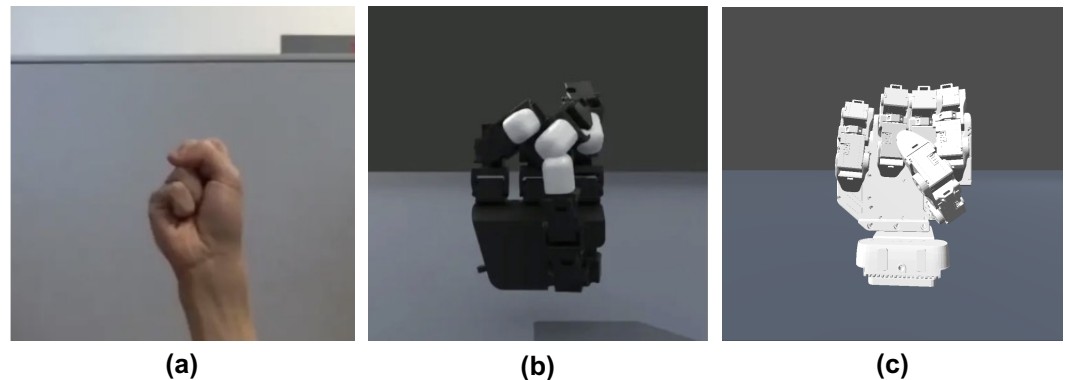

Figure 16: **Hand Retargeting Comparison**. (a) Human fist closure; (b) LEAP Hand with collisions and unnatural finger poses using [44]; (c) Retargeted RAPID Hand with natural configuration.

advancements in compact and powerful motors. This flexibility allows significant reductions in hand size without sacrificing dexterity.

**Expandable Perception Alignment.** The RAPID Hand's perception alignment framework integrates multi-modal data through temporal synchronization and spatial alignment, ensuring compatibility with diverse sensor types.

The framework supports the seamless integration of various tactile sensors on the fingertips, including optical, magnetic, and capacitive-based sensors. Additionally, it accommodates wrist-mounted cameras, such as fisheye or stereo cameras, and allows for further expansion with additional palm or pulp-mounted sensors and external vision systems. A key feature of this framework is its hardware-level synchronization, which ensures that all sensors—across different modalities—are precisely aligned in time. This prevents exposure mismatches between cameras, a common issue in software-only synchronization, which can lead to data inconsistency and degraded perception accuracy. By addressing these challenges at the hardware level, the RAPID Hand provides a reliable and adaptable perception system for complex manipulation tasks.

## A.2 Learning Dexterous Skills

To evaluate the RAPID Hand's performance and potential, we employ imitation learning on three challenging tasks: multi-fingered nonprehensile retrieval, object-in-hand translation, and rolling. This section details the data collection interface, task specifications, and our whole-hand visuotactile policy.

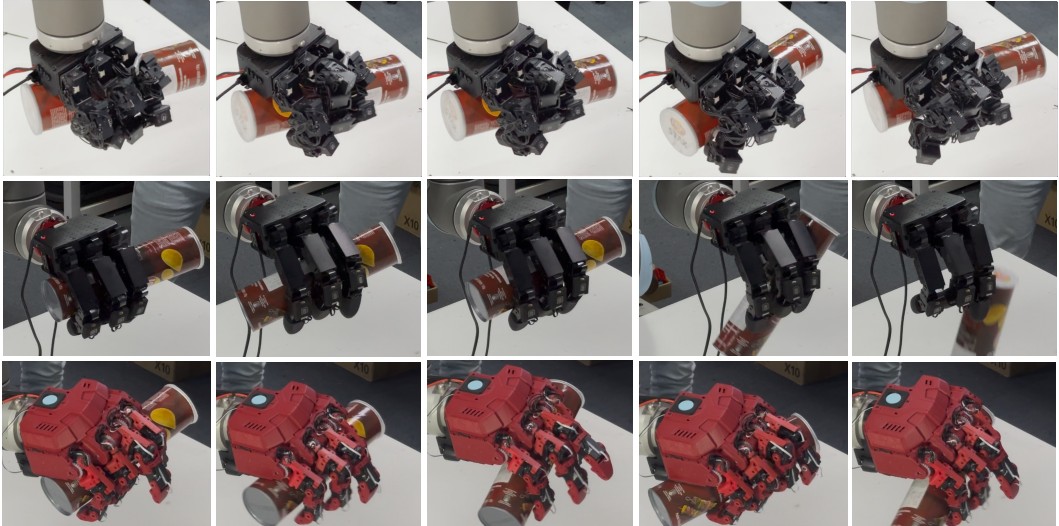

Figure 17: **Teleoperated In-Hand Translation Comparison.** Teleoperation results of in-hand object translation using (top) LEAP Hand, (middle) Allegro Hand, and (bottom) RAPID Hand. The RAPID Hand enables stable lateral translation with coordinated multi-finger motion, while LEAP and Allegro exhibit limited or unstable motions.

### A.2.1 Adaptive Robot Hand Motion Retargeting

To enable accurate and generalizable teleoperation of a multi-DoF robotic hand, we formulate a comprehensive retargeting objective that combines conformal-aligned constraints, contact-aware coupling constraints, and temporal smoothness. This formulation builds upon the structural similarity between the human and robotic hand by first calibrating and adjusting the human keypoints to spatially match the robot's geometry. However, aligning individual fingers alone is often insufficient for faithfully capturing complex behaviors such as pinching or coordinated grasping. To address this, we incorporate a multi-finger coordination term that dynamically enforces relative pose constraints based on inter-finger proximity. Finally, a temporal smoothness regularizer mitigates abrupt joint fluctuations, ensuring consistent and stable trajectories over time.

As shown in Eq. 1 The full retargeting objective is expressed as:

$$
\min_{q(t)} \lambda_1 \underbrace{\sum_{(i,j)\in\mathcal{K}} \left\| v_{i,j}(t) - \text{FK}_{i,j}(q(t)) \right\|^2}_{\text{Conformal-aligned Constraint}} + \lambda_2 \underbrace{\sum_{i\in\mathcal{I}} \omega_i(t) \left\| \Delta_i(t) - g_i(q(t)) \right\|^2}_{\text{Contact-aware Coupling Constraint}} + \lambda_3 \underbrace{\left\| q(t) - q(t-1) \right\|^2}_{\text{Temporal Smoothness}},
\tag{2}
$$

where $q(t) \in \mathbb{R}^n$ represents the joint angle vector of the RAPID Hand at time $t$.

**(i) Conformal-aligned Constraint** The first term minimizes spatial discrepancies between adjusted human hand keypoints and their corresponding positions on the robotic hand, where $\mathcal{K} \subseteq \{(i,j)|0 \leq i \leq 4, 0 \leq j \leq n_i\}$ defines pairs of keypoints for alignment, typically fingertip and intermediate joints and $v_{i,j}(t) \in \mathbb{R}^3$ denotes the adjusted position of the $j$-th keypoint on the $i$-th finger of the human hand at time $t$. The forward kinematics function $\text{FK}_{i,j}(q(t))$ computes the corresponding keypoint based on current joint angles $q(t)$.

To compute the adjusted keypoints $v_{i,j}(t)$, we first define the detected human hand keypoints as follows:

$$
\mathcal{V}_i = \{w_{i,0}, w_{i,1}, \ldots, w_{i,n_i}\}, \quad i = 0, \ldots, 4
\tag{3}
$$

where each keypoint $w_{i,j} \in \mathbb{R}^3$ represents the position of the $j$-th keypoint on the $i$-th finger, expressed in the wrist coordinate frame.

Human keypoint adjustments involve scaling each finger phalangeal segment and translating overall finger positions. Adjusted human keypoint positions, $v_{i,j}(t) \in \mathbb{R}^3$, are computed using calibration data:

- Scaling factors $r_{i,j}$ for each finger phalangeal segment:

$$r_{i,j} = \frac{\|\text{FK}_{i,j+1}(q_0) - \text{FK}_{i,j}(q_0)\|}{\|w^*_{i,j+1} - w^*_{i,j}\|} \tag{4}$$

where $\text{FK}_{i,j}(q_0)$ denotes the RAPID Hand keypoint at initial configuration $q_0$, and $w^*_{i,j}$ is the corresponding static calibrated human keypoint.

- Translational adjustments $u_i$ for each finger:

$$u_i = \text{FK}_{i,\text{mcp}}(q_0) - w^*_{i,\text{mcp}} \tag{5}$$

where $\text{FK}_{i,\text{mcp}}(q_0)$ and $w^*_{i,\text{mcp}}$ represent MCP joint positions of the robotic and human hand.

Adjusted keypoints are computed as:

$$v_{i,j} = \begin{cases} w_{i,0}, & j = 0 \\ v_{i,j-1} + r_{i,j-1}(w_{i,j} - w_{i,j-1}) + u_i, & j = 1 \\ v_{i,j-1} + r_{i,j-1}(w_{i,j} - w_{i,j-1}), & j \geq 2 \end{cases} \tag{6}$$

These adjusted keypoints preserve the relative orientations between finger phalangeal segments while scaling the overall hand geometry to match the robotic counterpart, thereby eliminating the need for introducing an additional global scaling factor $\alpha$ to manually align human hand keypoints.

**(ii) Contact-aware Coupling Constraint**   The second term ensures anthropomorphic consistency during grasping by enforcing relative positional constraints among fingers, particularly between the thumb and other fingers. This constraint is crucial for accurately capturing complex multi-finger interactions such as pinching and grasping. We first define the relative vector between the thumb and the fingertip of finger $i$ on the human hand as:

$$\Delta_i(t) = w_{i,\text{fingertip}}(t) - w_{\text{thumb,fingertip}}(t) \tag{7}$$

where $w_{i,\text{fingertip}}(t)$ and $w_{\text{thumb,fingertip}}(t)$ denote the positions of the fingertip of finger $i$ and the thumb, respectively, at time $t$.

Correspondingly, robotic relative vectors are calculated as:

$$g_i(q(t)) = \text{FK}_{i,\text{fingertip}}(q(t)) - \text{FK}_{\text{thumb,fingertip}}(q(t)) \tag{8}$$

To dynamically emphasize the coordination constraint based on the real-time interaction intensity between fingers, we introduce adaptive weighting factors $\omega_i(t)$. These weights are computed by assessing the normalized distance between the thumb and each finger:

$$d_i(t) = 1 - \frac{\|\Delta_i(t)\| - d_{\min,i}}{d_{\max,i} - d_{\min,i}} \tag{9}$$

where $d_{\min} \approx 0$ corresponds to finger-thumb contact, and $d_{\max}$ is derived from calibration data when the hand is fully extended. Consequently, $d_i(t) \in [0, 1]$ quantitatively represents the proximity of the thumb to finger $i$.

We apply a sigmoid function to smoothly adjust the influence of coordination:

$$\omega_i(t) = \frac{1}{1 + e^{-k(d_i(t) - c)}} \tag{10}$$

where $k$ controls the steepness and $c$ sets the sensitivity threshold of the sigmoid curve. Such adaptive modulation ensures that strong coordination constraints are enforced when fingers are close (e.g., during grasping) and significantly reduced when fingers are separated, thus preserving natural hand movements without unnecessary constraints.

**(iii) Temporal Smoothness**   The final term penalizes abrupt joint angle changes between successive time steps:

$$\|q(t) - q(t-1)\|^2 \tag{11}$$

Typically, $\lambda_1, \lambda_2, \lambda_3 = 1$ to equally balance optimization components.

The optimization is solved in real-time using Sequential Least-Squares Quadratic Programming (SLSQP) [62], providing stable and precise motion retargeting with minimal manual tuning. Leveraging pre-calibrated human keypoints ensures accuracy and scalability of teleoperation.

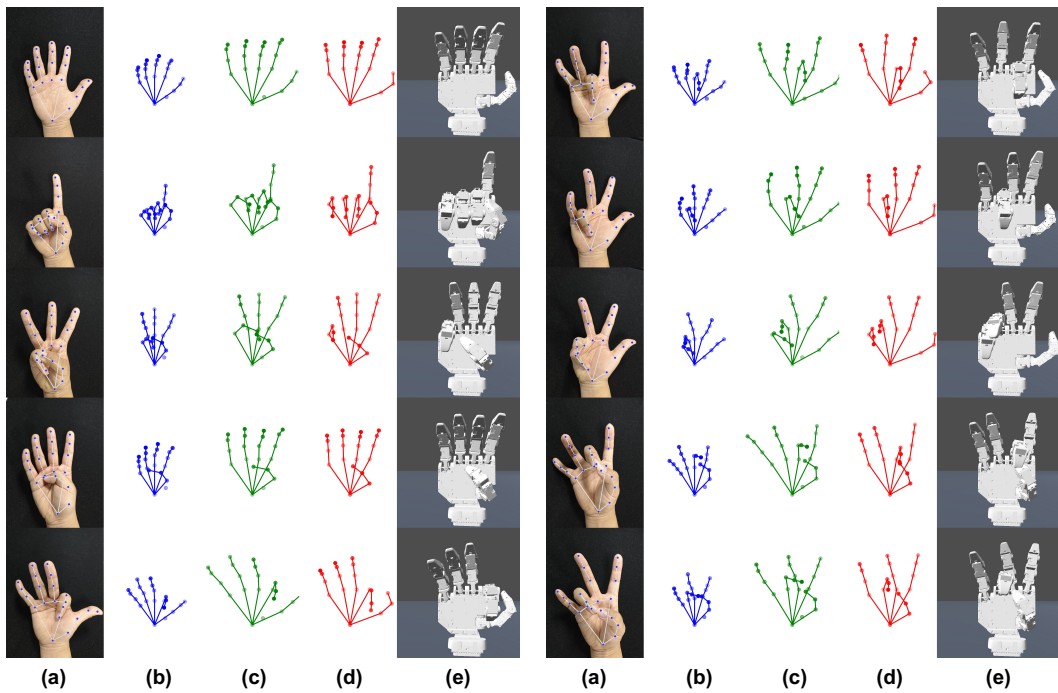

Figure 18: **Retargeting Processing Pipeline.** (a) hand images with keypoint annotations, (b) detected keypoints using MediaPipe [52], (c) adjusted keypoints after calibration using Eq. 6, (d) corresponding robot keypoints computed via optimization from Eq. 1, and (e) rendered robot hand poses after motion retargeting.

### A.2.2 Task Specifications

**Object-in-Hand Translation.** This task involves repositioning objects within the grasp to correct suboptimal initial poses, which is essential for tasks such as adjusting a bottle's position for pouring. A human operator provides a chip bottle to the robot at an arbitrary starting position. Using its five-fingered dexterity, the robot slides the bottle laterally to the leftmost edge of its grasp while maintaining a secure hold. Success is achieved only if the bottle reaches the target position without slipping or dropping.

**Object-in-hand Rolling.** A fundamental skill in food preparation (e.g., peeling vegetables), this task requires the robot to continuously roll cylindrical objects like corn, zucchini, or eggplant within its grasp. The robot must adapt to object shape, texture, and hardness variations while maintaining steady rotation. Successful execution ensures the object remains in motion without slipping or stalling.

**Multi-fingered Nonprehensile Retrieval.** Inspired by human dexterity in cluttered spaces, this task challenges the robot to retrieve a target object (e.g., from a drawer) obstructed by surrounding items. Using nonprehensile motions, the RAPID Hand first clears obstructions with its fingers before grasping the target. A successful retrieval requires completing both stages without failure.

### A.2.3 Learning Whole-Hand Visuotactile Policy

We train a whole-hand visuotactile policy using collected demonstrations to infer joint-space actions (e.g., 26-DoF sequences) from multi-modal observations. The policy architecture is based on Diffusion Policy [50], a generative model leveraging a time-series diffusion transformer to denoise and predict actions conditioned on historical observations, as explored in recent imitation learning works [63–65].

As shown in Fig. 19, at time step $t$, the input observations $\mathbf{O}_t = \{\mathbf{V}_t, \mathbf{T}_t, \mathbf{P}_t\}$ include:

- RGB vision $\mathbf{V}_t \in \mathbb{N}_0^{T_o \times 640 \times 480 \times 3}$: Wrist camera images over $T_o$ timesteps.

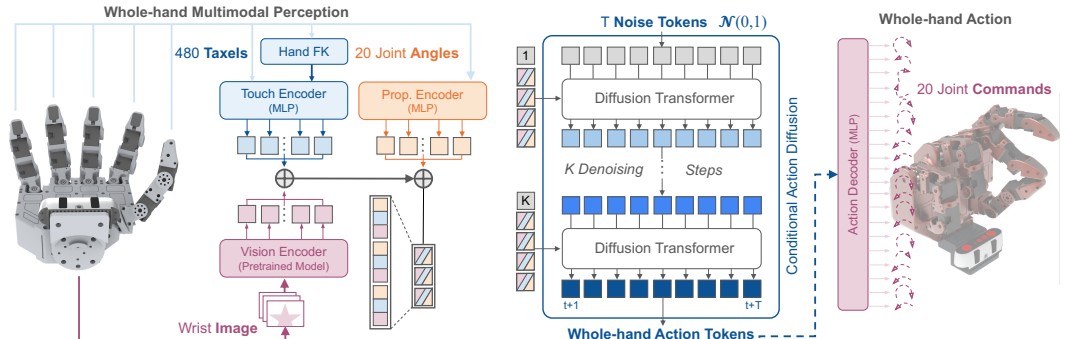

Figure 19: **Whole-hand Visuotactile Policy** leverages whole-hand perception for whole-hand dexterity. Left: The current robot and environment state is observed via RAPID Hand's wrist camera, fingertip touch, and its 20 joint angles, converted to local touch point cloud using hand forward kinematics. Each image (pink) is represented by the class token of a vision foundation model. Each tactile reading (blue) and the corresponding spatial information is embedded using our touch encoding. Twenty joint angles (orange) are embedded using a proprioceptive encoder. The concatenation of each camera's image, touch, and proprioception tokens yields a whole-hand multimodal perception token. Right: Our whole-body visuotactile policy consumes these vision, touch tokens, proprioception, and denoising-step tokens as conditions via cross-attention. We diffuse $T$ whole-hand dexterity tokens (blue), each corresponding to an action time step. Per time step, we project the predicted dexterity token to the 20 joint angles to be achieved by the dexterity action.

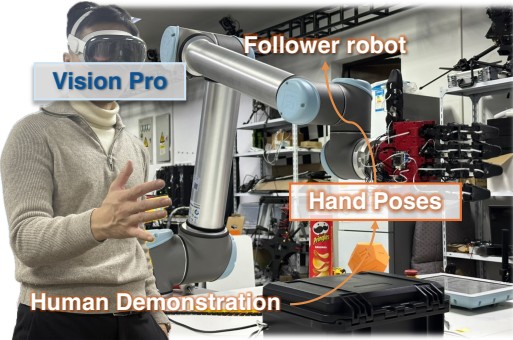

Figure 20: **Data Collection Interface.** The operator uses an Apple Vision Pro to control the robot, whose hand poses are sent to the robot in real-time as position targets. The human hand poses are recorded as target actions, while the images, tactile signals, and joint angles of the robotic hand and arm are recorded as observations.

- Touch $\mathbf{T}_t \in \mathbb{N}_0^{T_o \times 4 \times 12 \times 8 \times 5}$: Fingertip taxel (tactile pixel) readings from 5 tactile pads and the corresponding spatial position of each taxel calculated from hand's forward kinematics.
- Proprioception $\mathbf{P}_t \in \mathbb{R}^{T_o \times 26}$: Joint angles for the 20-DoF hand and the 6-DoF arm.

To extract in-hand visual features, we use a vision encoder initialized with a pre-trained ResNet-18. However, when objects occlude the robot's fingers during manipulation, tactile signals become critical—especially as finger movements dynamically and the contact state with the in-hand object is not accessible for vision. To enhance the policy's spatiotemporal awareness, we compute each taxel's spatial position in the hand frame using forward kinematics and current proprioception, requiring the perception of temporal synchronization and spatial alignment (Section 3.2). RGB images are extracted by ResNet-18 and compressed into 128-dimensional vectors by an MLP. Taxel readings and their positions are concatenated and embedded into a 64-dimensional vector via MLPs. Proprioception is similarly projected to a 192-dimensional vector. These embeddings are combined to form a 384-dimensional state representation, capturing the robot's whole-hand interaction with the environment at time $t$.

The policy generates action sequences $\mathbf{A}_t = \{a_{t+1}, \ldots, a_{t+T_a}\} \in \mathbb{R}^{T_a \times 26}$, which respecify 26-DoF joint targets (20 for the hand, 6 for the arm) over $T_a$ future timesteps. During training, denoising

iterations refine the noisy action proposals $\mathbf{A}_t^k$ (indexed by step $k$), which are processed as conditional inputs tokens in transformer decoder blocks. Observations $\mathbf{O}_t$ serve as conditional inputs via multi-head cross-attention layers, enabling the decoder to align sensorimotor data with action sequences. Cross-attention [66] integrates multi-modal observations by learning latent mappings between perception streams (visual, tactile, proprioceptive) and corresponding robotic actions, ensuring coordinated whole-hand manipulation. In our implementation, the policy is trained on an NVIDIA TITAN X GPU with a batch size of 32. We set the observation horizon $T_o = 1$, the action prediction horizon to $T_p = 64$, and the action execution horizon to $T_a = 64$. All policies are trained for 300 epochs and deployed at 10 Hz.

## A.3 Limitations

RAPID Hand still faces several practical limitations. Its overall size is constrained by the use of servo motors, which, while affordable and accessible, limit further miniaturization and restrict deployment in more compact robotic systems. Additionally, the current teleoperation interface lacks direct haptic feedback. This absence reduces the operator's ability to perceive subtle contact forces, increasing the risk of unintended excessive force during demonstrations. Future work will focus on integrating more compact, high-performance actuators and adding haptic feedback mechanisms to improve control precision, task safety, and user experience.

## A.4 Societal Impact

By open-sourcing RAPID Hand, we aim to provide a practical and accessible platform for dexterous manipulation research. The system's low cost and modular design allow broader adoption, enabling more researchers to explore in-hand manipulation without relying on expensive or proprietary hardware. We hope this will help accelerate progress in generalist robot autonomy. At the same time, care must be taken to ensure responsible use, particularly in safety-critical or human-facing applications.

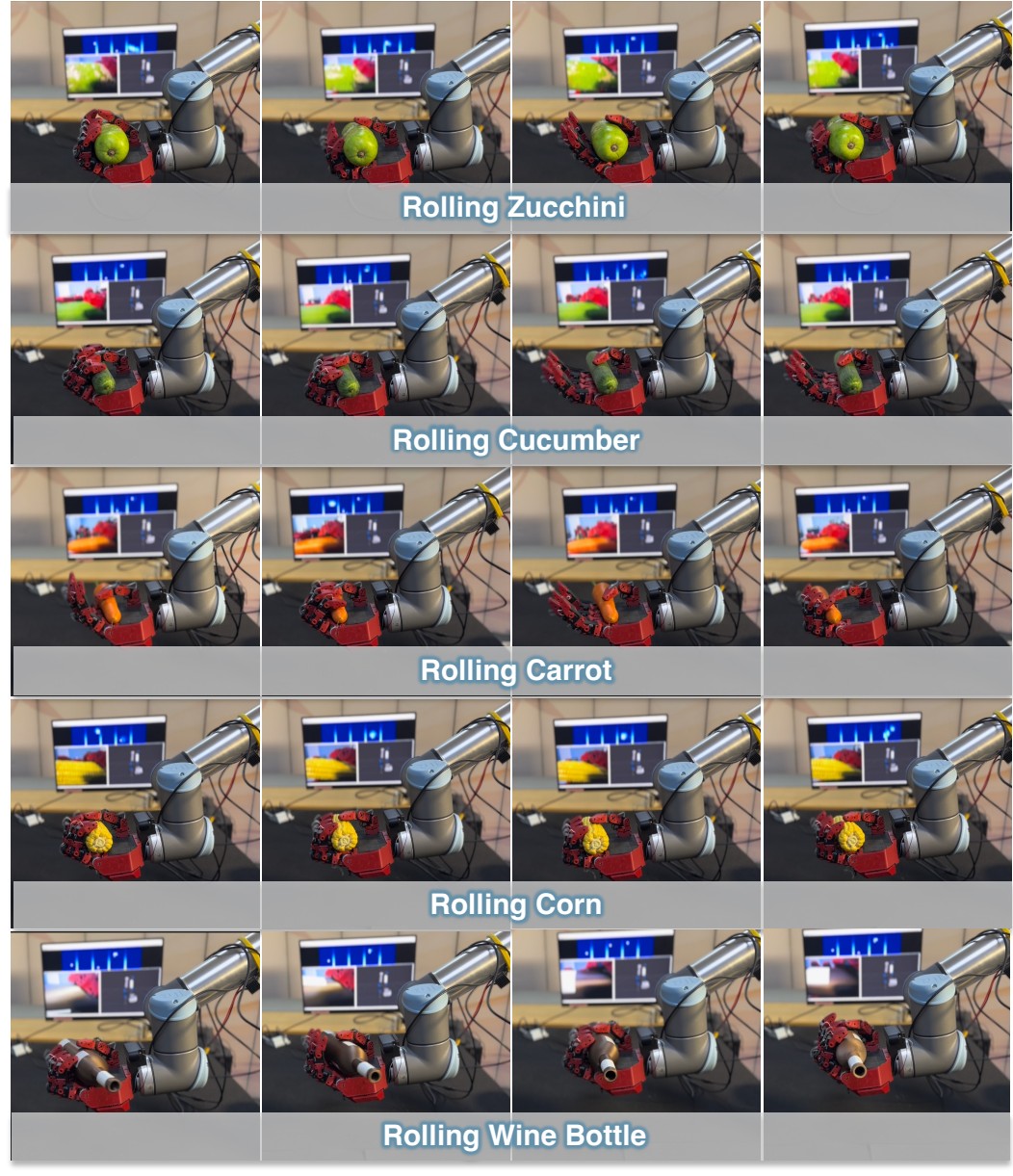

Figure 21: **Generalization Performance Visualization of In-hand Rolling.**

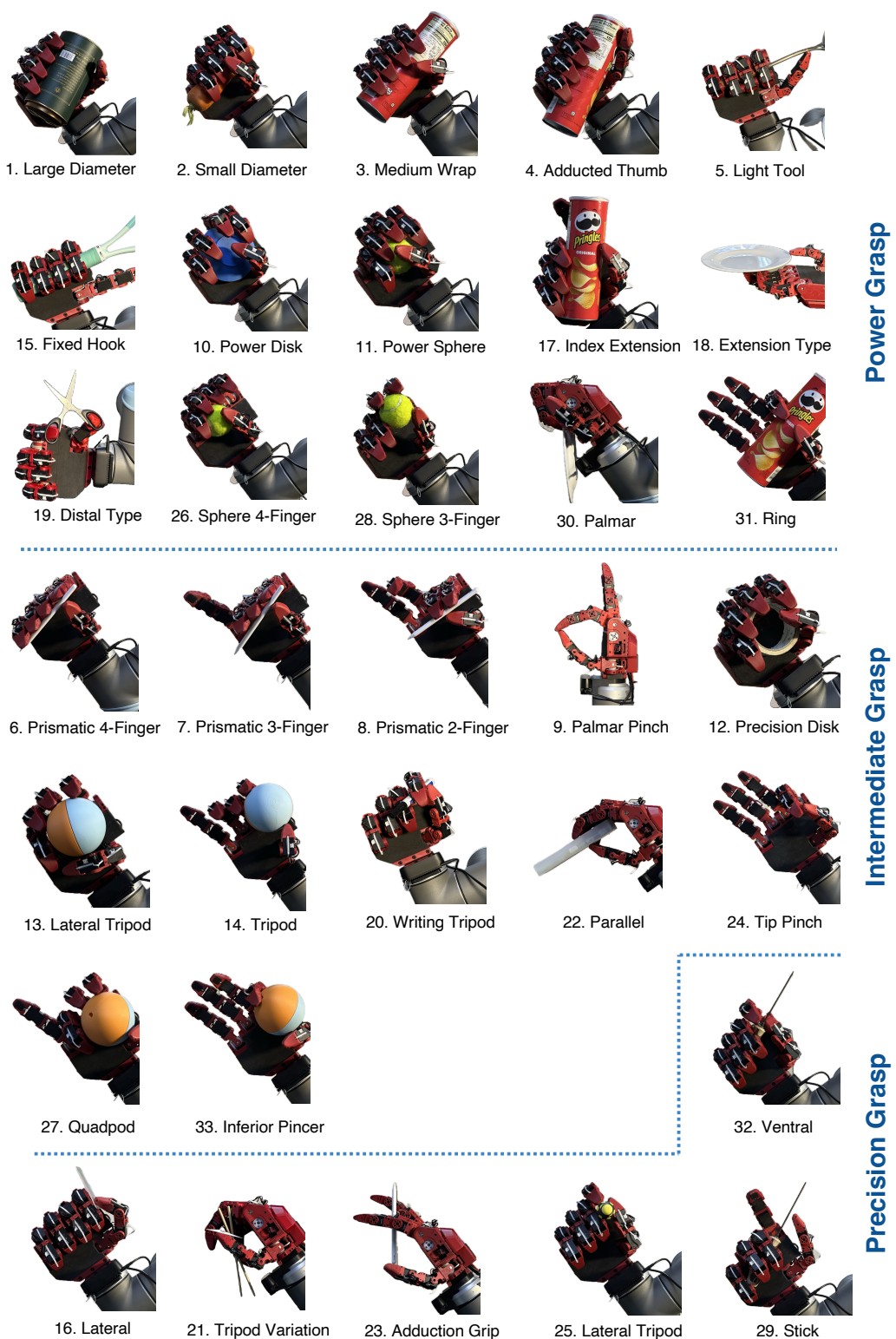

Figure 22: **Grasp Taxonomy of the RAPID Hand.** Grasp poses and their indices correspond to the Feix taxonomy [51].

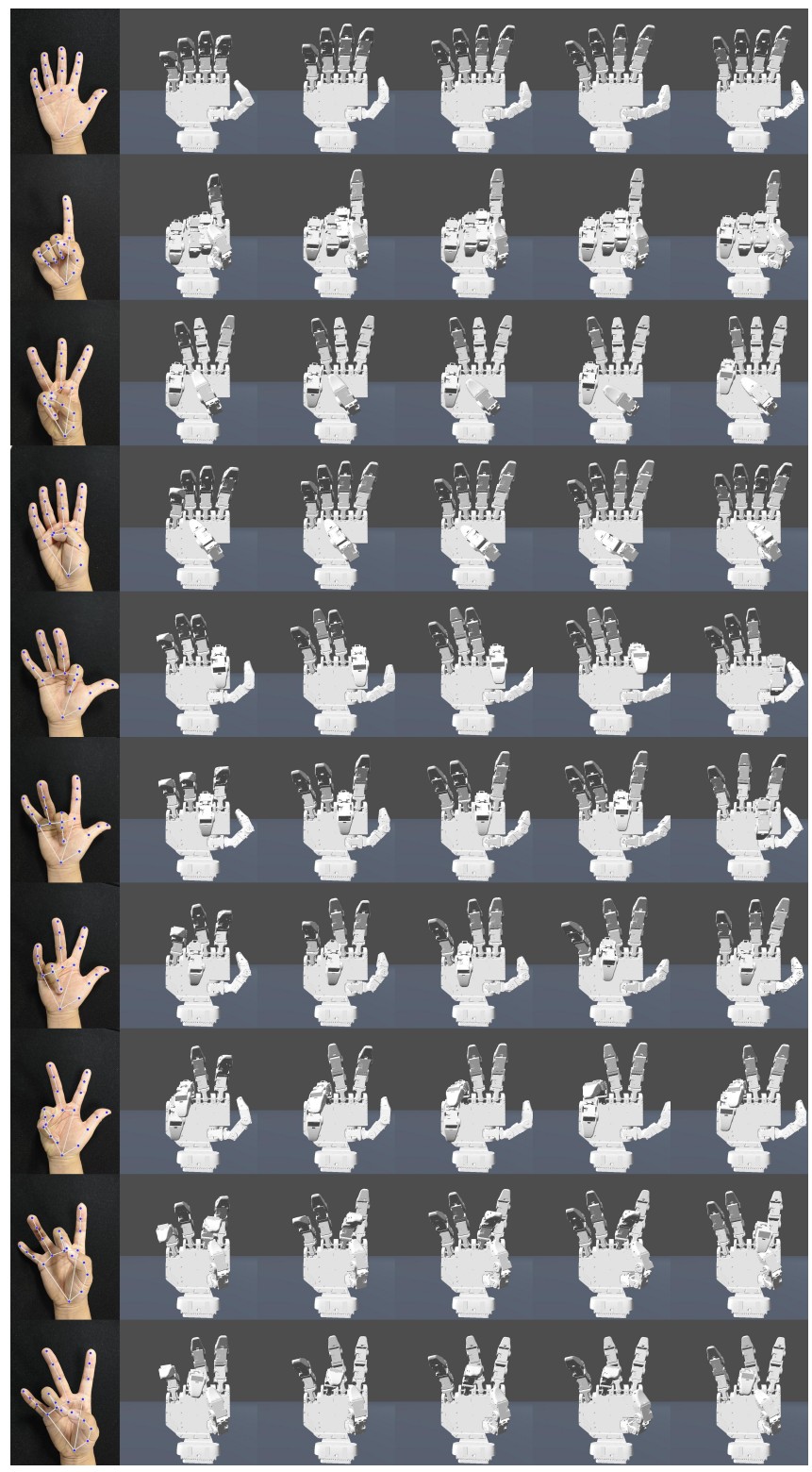

Figure 23: **Qualitative Comparison of Retargeting Results on RAPID Hand.** From left to right: (1) original human hand gestures, (2–5) results of baseline method [44] using global scaling factors $\alpha = 1.25, 1.50, 1.75, 2.00$, and (6) results from our proposed method.

