# OpenReview forum: "RAPID Hand: Robust, Affordable, Perception-Integrated, Dexterous Manipulation Platform for Embodied Intelligence"
_NeurIPS.cc/2025/Conference — NeurIPS 2025 poster_

### Official Review · Reviewer_65oh · 2025-06-07

**Clarity:** 3
**Significance:** 3
**Originality:** 3
**Rating:** 5
**Confidence:** 4

**Summary:**

This paper presents RAPID Hand, a low-cost, modular, and open-source 20-DoF robotic hand platform designed for collecting high-quality real-world dexterous manipulation data. The system integrates whole-hand multimodal perception—including vision, touch, and proprioception—with sub-7ms synchronization. It allows researchers to collect data with high-DoF teleoperation and high-fidelity perception. The effectiveness of the system has been validated by collecting demonstrations on three common manipulation tasks and training a diffusion policy to achieve stably high success rate.

**Questions:**

1. In terms of labor, time, and expertise, what does it take to assemble a RAPID Hand?
2. How robust is the hand? How often does it break or burn out? How does the system handle sensor degradation or calibration drift over time? The paper shows stable tracking under load, no overheating, and larger load capacity over LEAP Hand. However, failure rates or long-term durability (e.g., motor burnout, sensor degradation) are not reported.
3. Can the hand be equipped on mainstream robot arms and bodies? In the paper, the hand is equipped on UR10e. But it does not discuss integration with other arms or humanoid platforms, nor does it provide mounting specs or interface standards.
4. Can the system fully replay a recorded trajectory?
5. How transferable is the data or policies on other widely used dexterous hands (e.g., Allegro, Shadow) to RAPID Hand?

**Ethical Concerns:**

["NO or VERY MINOR ethics concerns only"]

**Final Justification:**

Taking the authors' rebuttal into consideration, RAPID Hand presents sufficient evidence of being robust, reliable, and economical. The authors also have a plan for future support and the development of RAPID Glove, which holds great potential for widespread adoption by the community. Therefore, I have decided to change my rating to "Accept."

**Limitations:**

1. The paper does not mention whether the system can reliably replay recorded trajectories, which is important for assessing control fidelity, calibration stability, and reproducibility of demonstrations.
2. While the RAPID Hand is cost-effective in terms of hardware components, its assembly requires time, technical expertise, and access to tools such as 3D printers. These practical barriers may limit its accessibility to broader segments of the research community, particularly those without prior experience in hardware prototyping.
3. Given the number of existing dexterous hands that have been in used in the community for years, one general question for a new hardware system is how easily it can be integrated with mainstream robotic arms or humanoid platforms. Furthermore, it remains unclear whether data or policies trained on other widely used dexterous hands (e.g., Allegro, Shadow) can be effectively transferred to RAPID Hand, which is a critical consideration for adoption in existing research pipelines.

**Paper Formatting Concerns:**

None.

**Quality:**

3

**Strengths And Weaknesses:**

Strengths:
1. RAPID Hand has 20 DoF with anthropomorphic kinematics that support highly synchronized teleoperation and dexterous manipulation.
2. The paper lays out a comprehensive list of critical hardware specifications, such as thumb opposability volume, fingertip force, weight, size, joint accuracy, record frequency, robustness under latency and dropout.
3. RAPID Hand uses low-cost and off-the-shelf components with a total cost of only $3,500 and is thus more repairable.
4. RAPID Hand collects three modalities—vision from a wrist-mounted RGB-D camera, touch from piezoresistive tactile sensors, and 20-DoF proprioception from the joints—that enable high performance on common manipulation tasks.
5. The paper shows that a diffusion policy trained on the data collected by this system can achieve stably high success rate on three different common manipulation tasks.

Weaknesses:
1. As RAPID Hand requires manual assembly, even if the bill of materials is low, some hidden costs, such as time, tooling (e.g., 3D printer), and expertise required for assembly are non-trivial and may limit accessibility for some labs.
2. The paper does not mention repeatability by replaying collected trajectories, which should be a straightforward evaluation of the accuracy of the entire system.
3. While the paper emphasizes the utility of wrist-mounted vision for in-hand perception, it does not discuss the compatibility of RAPID Hand with diverse robot arms or full-body humanoid platforms. As dexterous hands are increasingly deployed in integrated systems, the lack of evaluation or guidance on cross-platform adaptability may limit the system's practical adoption.

---

> ### Author Rebuttal · Authors · 2025-07-30
>
> We thank the reviewer for the detailed reading of our paper and constructive suggestions! We hope our responses adequately address the following questions about our work. Please let us know if there’s anything we can clarify further.
>
> ```
> 1. As RAPID Hand requires manual assembly, even if the bill of materials is low, some hidden costs, such as time, tooling (e.g., 3D printer), and expertise required for assembly are non-trivial and may limit accessibility for some labs. In terms of labor, time, and expertise, what does it take to assemble a RAPID Hand?
> ```
> **Reply**: Manual assembly inevitably involves non-trivial costs in time, tooling, and expertise beyond the bill of materials, a challenge shared by other hardware platforms such as LEAP Hand (about 4 hours to assemble [1]). In our experience, assembling a RAPID Hand takes about 6 hours for an undergraduate student familiar with standard prototyping tools. Although RAPID Hand includes slightly more components than LEAP Hand, the overall assembly difficulty is comparable. The required equipment is limited to a desktop 3D printer and common tools such as soldering kits and screwdrivers.
>
> To lower the barrier, we design the hand with modular components and provide detailed open-source CAD files, wiring diagrams, and assembly guides. While some effort is unavoidable, we believe the overall accessibility remains significantly higher than that of existing dexterous hands. To further lower the entry barrier, we plan to release step-by-step assembly instructions and partially assembled kits in future updates.
>
> ```
> 2. The paper does not mention repeatability by replaying collected trajectories, which should be a straightforward evaluation of the accuracy of the entire system. Can the system fully replay a recorded trajectory?
> ```
> **Reply**: Yes, the system can fully replay a recorded trajectory. Repeatability involves both the arm and the hand. For the arm, we use the widely adopted UR10e, which offers a pose repeatability of ±0.05 mm [2]. For the hand, we reported joint precision during motion (Fig. 8) and performance under varying loads (Fig. 9) in the appendix, both demonstrating consistent behavior. Moreover, RAPID Hand has autonomously executed repeated public demonstrations, further supporting the system’s repeatability.
>
> ```
> 3. While the paper emphasizes the utility of wrist-mounted vision for in-hand perception, it does not discuss the compatibility of RAPID Hand with diverse robot arms or full-body humanoid platforms. As dexterous hands are increasingly deployed in integrated systems, the lack of evaluation or guidance on cross-platform adaptability may limit the system's practical adoption. Can the hand be equipped on mainstream robot arms and bodies? In the paper, the hand is equipped on UR10e. But it does not discuss integration with other arms or humanoid platforms, nor does it provide mounting specs or interface standards.
> ```
> **Reply**:  RAPID Hand can be mounted on other robot arms and humanoid platforms, though some additional adapters are required. The hand uses a modular wrist structure in which the camera mount remains fixed, and only the wrist flange needs to be replaced when adapting to a different arm. Both the flange and the camera bracket follow standard mechanical interfaces, which facilitates integration across platforms.
>
> We are preparing wrist flanges for mainstream collaborative arms (e.g., Franka, xArm) and humanoid platforms (e.g., Unitree G1), together with detailed mounting specifications and interface standards to further support cross-platform use.
>
>
>
> ```
> 4. How robust is the hand? How often does it break or burn out? How does the system handle sensor degradation or calibration drift over time? The paper shows stable tracking under load, no overheating, and larger load capacity over LEAP Hand. However, failure rates or long-term durability (e.g., motor burnout, sensor degradation) are not reported.
> ```
> **Reply**: Based on our experience, RAPID Hand has remained generally robust under daily use for about six months. The only notable incident occurred when the arm lost control and the hand struck the table, breaking a finger linkage. The part was replaced within about 5 minutes, and the system continued to operate normally. In tasks such as multi-finger retrieval, where fingers frequently interact with stacked objects in confined spaces, the hand has so far functioned reliably without structural failures.
>
> We have not observed clear signs of degradation in the tactile sensors to date. For proprioception, we use a custom calibration board to reset each phalange, typically once per day, to maintain accuracy. Current-based position control also helps reduce the risk of motor burnout from stall-induced overheating. We will include these details in the paper to provide a clearer picture of the system’s robustness and durability.
>
>
>
>
> ```
> 5. How transferable is the data or policies on other widely used dexterous hands (e.g., Allegro, Shadow) to RAPID Hand?
> ```
> **Reply**: Cross-hand transfer remains a significant challenge in the field. Differences in actuation and transmission mechanisms, hand size, and sensing modalities (e.g., optical, piezoresistive, capacitive tactile sensors) all contribute to distributional gaps that affect transferability. For example, the Allegro hand lacks tactile sensing, while the Shadow hand is tendon-driven, making direct policy transfer difficult.
>
> A practical direction we see is to use data from other hands (e.g., Allegro, Shadow) as part of training robot foundation models [3, 4, 5], and then fine-tune with a smaller set of RAPID Hand demonstrations. While we have not yet conducted such experiments, we consider this an important avenue for future work.
>
> [1] Shaw, Kenneth, Ananye Agarwal, and Deepak Pathak. "Leap hand: Low-cost, efficient, and anthropomorphic hand for robot learning." Robotics: Science and Systems (RSS 2023).
>
> [2] https://www.universal-robots.com/media/1807466/ur10e_e-series_datasheets_web.pdf
>
> [3] Physical Intelligence, et al. “π0.5: a vision-language-action model with open-world generalization.” arXiv preprint arXiv:2504.16054, 2025.
>
> [4] Gemini Robotics Team, et al. “Gemini robotics: Bringing ai into the physical world.” arXiv preprint arXiv:2503.20020, 2025.
>
> [5] Johan Bjorck, et al. “GR00T-N1: An open foundation model for generalist humanoid robots.” arXiv preprint arXiv:2503.14734, 2025.

---

> > ### Comment · Reviewer_65oh · 2025-08-02
> >
> > Thank you for your clarifications.
> >
> > 1. For the assembly process, a tutorial video would be especially appreciated.
> > 2. For cross-platform integration, it is great to hear that mainstream platforms will be supported.
> > 3. "Based on our experience, RAPID Hand has remained generally robust under daily use for about six months." How intensive is the daily use?
> > 4. Your perspective on cross-hand transfer is appreciated. I still have slight concerns about the broader reception of the new hand considering the existing large volume of prior work and datasets built around other established dexterous hands. Without clear pathways, RAPID Hand may face limited adoption, which could lead to a negative feedback loop—lower usage resulting in reduced visibility and community support over time.

---

> > > ### Author Response · Authors · 2025-08-03
> > >
> > > Thanks for the reviewer’s feedback. We are preparing a step-by-step tutorial video for future updates. During data collection and experiments, RAPID Hand is used over 12 hours per day; afterward, it continues in public demonstrations, typically about once a week, depending on visitor schedules.
> > >
> > > RAPID Hand integrates compact hardware and synchronized multimodal sensing to support dexterous data collection. The bevel gear differential enables 20 mm finger thickness with robustness, and hardware-level synchronization achieves sub‑7 ms latency with fingertip tactile signals aligned into local “touch point clouds.” These features yield practical benefits: for example, in in-hand translation, RAPID Hand achieves smooth lateral motion, unlike Allegro, which often drops the object, or LEAP, which shows minimal movement.
> > >
> > > At the same time, RAPID Hand is similar to LEAP and Allegro in actuation, size, and cost, making transfer or adaptation from data collected with these platforms easier than with tendon-driven hands such as Shadow.
> > >
> > > Looking forward, we are developing **RAPID Glove**, a high-**R**esolution, **A**ffordable, ha**P**tic **I**nterface that, together with RAPID Hand, forms a complete, low-cost platform for collecting and open‑sourcing larger amounts of high-quality, contact-rich demonstrations.

---

> > > > ### Author Response · Authors · 2025-08-07
> > > > **Please let us know whether whether we address all the questions**
> > > >
> > > > Dear reviewer,
> > > >
> > > > We appreciate your reviews and comments. We hope our responses address your questions.
> > > >
> > > > As the deadline is only one day away, please let us know if you have further questions after reading our rebuttal.
> > > >
> > > > We hope to address all the potential issues during the discussion period.
> > > >
> > > > Thank you

---

> > > > > ### Comment · Area_Chair_SBKr · 2025-08-08
> > > > > **Please respond to the author comments**
> > > > >
> > > > > Hi, Please respond to the author comments. They have provided a detail update and it'd be good to provide them constructive feedback.

---

### Official Review · Reviewer_CUJk · 2025-06-28

**Clarity:** 4
**Significance:** 4
**Originality:** 4
**Rating:** 5
**Confidence:** 4

**Summary:**

This paper presents RAPID Hand, a 20-DoF robotic hand platform designed for collecting high-quality demonstration data for dexterous manipulation. The key contributions include: (1) a compact hand ontology with universal multi-phalangeal actuation achieving 20mm finger thickness, (2) a hardware-level perception framework integrating wrist vision, fingertip tactile sensing, and proprioception with sub-7ms synchronization, and (3) a high-DoF teleoperation interface with conformal-aligned and contact-aware constraints. The authors validate their system on three manipulation tasks (in-hand translation, rolling, and multi-fingered retrieval) using diffusion policy learning, demonstrating superior performance compared to prior works.

**Questions:**

1. Teleoperation latency: What is the end-to-end latency from human hand motion to robot execution? How does the optimization solver perform under rapid movements or when approaching joint limits?
2. Tactile sensor durability: Given that flat piezoresistive sensors are "inevitable to wear and tear," what is the expected lifetime under typical manipulation loads? How does sensor degradation affect policy performance?
3. Force control capabilities: The system uses position control throughout. How does this limit performance in tasks requiring precise force regulation, such as fragile object handling or assembly?

**Ethical Concerns:**

["NO or VERY MINOR ethics concerns only"]

**Quality:**

4

**Strengths And Weaknesses:**

## Strengths
1. Comprehensive hardware-software co-design: The paper presents a well-integrated system where hardware design, perception integration, and teleoperation interface are jointly optimized. This holistic approach addresses real practical challenges in dexterous manipulation data collection.
2. Innovative mechanical design: The universal multi-phalangeal actuation scheme with bevel gear differential mechanism is clever, achieving compact 20mm finger thickness while maintaining robustness. The parallel MCP joint design shows 2.3x better load capacity than LEAP Hand's tandem configuration.
3. Robust perception integration: The hardware-level synchronization achieving sub-7ms latency across modalities is impressive. The spatial alignment of tactile readings into "local touch point clouds" provides direct geometric correspondence, addressing a key limitation in prior tactile manipulation work.

## Weaknesses
1. Limited task complexity: While the three evaluated tasks demonstrate basic dexterous capabilities, they are relatively simple compared to real-world manipulation needs. Tasks like tool use, bimanual coordination, or fine assembly would better demonstrate the platform's capabilities for "generalist robot autonomy."
2. Perception limitations: The flat tactile sensors provide only normal force measurements, lacking shear force sensing crucial for slip detection and fine manipulation. The single wrist-mounted camera may suffer from occlusions during complex manipulations.

---

> ### Author Rebuttal · Authors · 2025-07-30
>
> We thank the reviewer for the detailed reading of our paper and constructive suggestions! We hope our responses adequately address the following questions about our work. Please let us know if there’s anything we can clarify further.
> ```
> 1. Limited task complexity: While the three evaluated tasks demonstrate basic dexterous capabilities, they are relatively simple compared to real-world manipulation needs. Tasks like tool use, bimanual coordination, or fine assembly would better demonstrate the platform's capabilities for "generalist robot autonomy."
> ```
> **Reply**: The three evaluated tasks, namely in-hand rolling, translation, and multi-finger retrieval, are chosen as atomic skills that test durability, sensing consistency, and dexterous control. More complex tasks such as tool use can be viewed as compositions of these fundamentals. RAPID Hand is designed as an affordable, reliable platform for consistent multimodal data collection. While advancing to more complex tasks would further demonstrate its potential, progress is constrained by the difficulty of collecting safe, high-quality demonstrations, since human teleoperation without tactile feedback makes producing dexterous motions on a different embodiment particularly challenging [1]. As a next step, we are exploring **RAPID Glove**, a high-**R**esolution, **A**ffordable, ha**P**tic **I**nterface for **D**emonstrating contact-rich manipulation, as an important direction for extending the platform’s capabilities.
>
> ```
> 2. Perception limitations: The flat tactile sensors provide only normal force measurements, lacking shear force sensing crucial for slip detection and fine manipulation. The single wrist-mounted camera may suffer from occlusions during complex manipulations.
> ```
> **Reply**: The current system uses flat fingertip tactile arrays to balance affordability with consistent performance. While shear force sensing can improve slip detection and fine manipulation, our tests with sensors from Paxini [2], TashanTec [3], and Xela Robotics [4] confirmed such benefits but also showed significant increases in cost, maintenance effort, and difficulty in ensuring high-resolution, consistent readings. Given our focus on cost-effectiveness and robustness, we chose flat arrays for this design.
>
> For vision, we recognize that a single wrist-mounted camera may lead to occlusions. Our perception framework can readily support hardware-synchronized external cameras, and we plan to explore this extension in future work to improve coverage during complex manipulations.
>
> ```
> 3. Teleoperation latency: What is the end-to-end latency from human hand motion to robot execution? How does the optimization solver perform under rapid movements or when approaching joint limits?
> ```
> **Reply**: The end-to-end latency from human hand motion to robot execution is mainly due to the UR collaborative arm, typically around 1–2 s. The RAPID Hand itself responds with minimal delay. While this latency is noticeable, teleoperators generally adapt after some practice, allowing us to collect demonstrations reliably.
>
> For rapid movements, the optimization solver can compute inverse kinematics solutions, but execution remains limited by the arm’s delay. To reduce risks of collision or hardware strain, we avoid aggressive motions and prevent the arm from approaching joint limits. We acknowledge that reducing arm latency would further improve teleoperation, and this could be addressed by adopting a robotic arm with faster execution in future work.
>
> ```
> 4. Tactile sensor durability: Given that flat piezoresistive sensors are "inevitable to wear and tear," what is the expected lifetime under typical manipulation loads? How does sensor degradation affect policy performance?
> ```
> **Reply**: Given that piezoresistive sensors are inevitably subject to wear and tear, we believe their durability is largely determined by fabrication processes and quality. For reference, we previously used the tactile sensor in [5], which maintained stable performance for about three months of daily manipulation before gradually declining. The sensors used in this work, according to supplier testing, can withstand over one million impacts at a 10 kg load. Based on our experience, since their deployment in November 2024, we have not needed to replace any, and no noticeable degradation has been observed.
>
> If replacement is required, each sensor is mass-produced and costs about **$70** [6], in contrast to alternatives such as Paxini [2] (**$400**) and Xela USkin [4] (**$2,000**), which helps keep maintenance affordable. To date, RAPID Hand has consistently performed in-hand translation and rolling tasks, including repeated public demonstrations, without evident loss of performance.
>
> ```
> 5. Force control capabilities: The system uses position control throughout. How does this limit performance in tasks requiring precise force regulation, such as fragile object handling or assembly?
> ```
>
> **Reply**: Using position control alone has inherent limitations, especially for tasks that require precise force regulation. In our evaluations, we occasionally observed excessive forces, such as minor surface damage to vegetables during in-hand rolling. In this work, we adopt position control to validate the stability and effectiveness of the RAPID Hand platform for autonomous policy learning, as it allows reliable and consistent execution across a range of contact-rich tasks, though we note that force control will be essential for more delicate tasks such as fragile object handling or fine assembly and plan to pursue this in future work.
>
> [1] Yin, Zhao-Heng, et al. "DexterityGen: Foundation controller for unprecedented dexterity." Robotics: Science and Systems (RSS 2025).
>
> [2] https://paxini.com/
>
> [3] https://www.tashantec.com/
>
> [4] https://www.xelarobotics.com/tactile-sensors
>
> [5] Lu, Peng, et al. "Thermoformed electronic skins for conformal tactile sensor arrays." 2024 IEEE International Conference on Robotics and Automation (ICRA 2024).
>
> [6] https://www.moxiantech.com/robot-scene/14

---

> > ### Author Response · Authors · 2025-08-07
> > **Please let us know whether we address all the issues**
> >
> > Dear Reviewer
> >
> > We appreciate your reviews and comments. We hope our responses address your questions.
> >
> > As the deadline is only one day away, please let us know if you have further questions after reading our rebuttal.
> >
> > We hope to address all the potential issues during the discussion period.
> >
> > Thank you

---

> > > ### Comment · Area_Chair_SBKr · 2025-08-08
> > > **Please respond to the author comments**
> > >
> > > Hi, Please respond to the author comments. They have provided a detail update and it'd be good to provide them constructive feedback.

---

### Official Review · Reviewer_GKwW · 2025-06-30

**Clarity:** 3
**Significance:** 2
**Originality:** 3
**Rating:** 4
**Confidence:** 5

**Summary:**

This paper presents RAPID Hand, a low-cost, 20-DoF robotic hand with integrated multimodal perception for dexterous manipulation. The platform is validated through multiple experiments. By prioritizing affordability and open sourcing the hardware, software, the authors aim to make dexterous manipulation research more accessible and scalable.

**Questions:**

Questions:
1.	In Figure 2: Why does the paper not compare the proposed hand with the five-finger Shadow Hand? This omission appears significant.
2.	In Figure 4: What is the server being used? If it is running on typical operating systems like Windows or Linux, which are not real-time systems, the reported latency values seem unconvincing.
3.	In Figure 7: What does the bar represent—mean values? If so, where are the error bars or other indicators to show the distribution? Further, a significance test should be considered to support the conclusions drawn from the results.

**Ethical Concerns:**

["NO or VERY MINOR ethics concerns only"]

**Limitations:**

Limitations:
1.	The main text focuses primarily on the platform’s characterization, with limited discussion on its practical applications.
2.	The unique advantages and contributions of the platform to the whole AI community are not adequately presented or verified.

**Quality:**

3

**Strengths And Weaknesses:**

Strengths:
1.	The paper introduces a well-designed hardware platform with a high degree of freedom, which is directly and exactly actuated.
2.	The integration of hard-synchronized multimodal perception enhances the efficiency and quality of data collection.

Weaknesses:
1.	The paper emphasizes the characterization of the hardware system but devotes much less attention to the unique capabilities and advantages that the new hardware offers. This imbalance may reduce its appeal to the broader NeurIPS audience.
2.	Certain critical aspects, such as system latency, are not fully validated.
3.	The advancements introduced by this new hardware are not sufficiently evaluated or contextualized.

---

> ### Author Rebuttal · Authors · 2025-07-30
>
> We thank the reviewer for the detailed reading of our paper and constructive suggestions! We hope our responses adequately address the following questions raised about our work. Please let us know if there’s anything we can clarify further.
>
> ```
> 1. The paper emphasizes the characterization of the hardware system but devotes much less attention to the unique capabilities and advantages that the new hardware offers. This imbalance may reduce its appeal to the broader NeurIPS audience.
> ```
> **Reply**: We conclude the unique capabilities of RAPID Hand as follows.
>
> **First**, the hardware is co-designed with perception and teleoperation to address practical challenges in dexterous manipulation data collection. The universal multi-phalangeal actuation scheme with a bevel gear differential achieves a compact 20 mm finger thickness while maintaining robustness, and the parallel MCP joint provides 2.3× higher load capacity than the LEAP Hand’s tandem design.
>
> **Second**, on the sensing side, hardware-level synchronization ensures sub-7 ms multimodal latency, and fingertip tactile signals are spatially aligned into local “touch point clouds,” enabling direct geometric correspondence not available in prior tactile systems.
>
> **Third**, these capabilities translate into practical benefits: in in-hand translation, RAPID Hand achieves smooth lateral finger motion, while Allegro often drops the object and LEAP shows minimal movement. Compared with prior work [1], our setup also removes simplified assumptions such as fixed end-effectors and table support, as shown in the supplementary video.
>
> We will devote more attention to highlighting these unique capabilities in the final version.
> ```
> 2. Certain critical aspects, such as system latency, are not fully validated. Particularly, in Figure 4: What is the server being used? If it is running on typical operating systems like Windows or Linux, which are not real-time systems, the reported latency values seem unconvincing.
> ```
> **Reply**: The sub-7 ms latency reported in Figure 4 refers specifically to multimodal synchronization, not end-to-end teleoperation delay. It measures the interval between the MCU (Figure 4 “Hard Sync”) receiving fingertip haptic data and triggering the camera exposure. This was verified using a logic analyzer.
>
> Such low latency is achieved through hardware-level synchronization (e.g., MCU-based triggering and timestamp alignment) and is independent of the upper-level operating system. While Linux is not a real-time OS, it does not affect this measurement, as the synchronization occurs entirely on the MCU. We will clarify this definition in the final version.
>
>
>
> ```
> 3. In Figure 2: Why does the paper not compare the proposed hand with the five-finger Shadow Hand? This omission appears significant.
> ```
> **Reply**: We compare RAPID Hand with Allegro and LEAP because they are fully actuated, direct-drive, relatively affordable, and widely adopted in recent works [3, 4, 5], making them the most relevant baselines. The Shadow Hand, while highly capable, follows a design focus different from ours, being tendon-driven and oriented toward more advanced manipulation skills. Our work emphasizes platforms suited for long-term autonomous operation and large-scale data collection, where accessibility and robustness are critical. While Shadow is not included in Figure 2, we provide a specification-level comparison in Table 2 and plan to add a size comparison in the final version.
>
>
>
>
>
> ```
> 4. In Figure 7: What does the bar represent—mean values? If so, where are the error bars or other indicators to show the distribution? Further, a significance test should be considered to support the conclusions drawn from the results.
> ```
> **Reply**: **Mean Values and Standard Deviations**
> The bars in Figure 7 show the mean action MSE for each condition. The mean and standard deviation for each setting are summarized below:
>
> | Task | Condition | Mean | Std |
> |-------------|---------------|-------|--------|
> | Rolling | Ours | 0.84 | 0.0239 |
> | Rolling | w. latency | 0.93 | 0.0277 |
> | Translation | Ours | 0.98 | 0.0809 |
> | Translation | w. latency | 1.16 | 0.1152 |
> | Retrieval | Ours | 0.51 | 0.0523 |
> | Retrieval | w. latency | 0.56 | 0.0554 |
>
> **Statistical Significance**
> We conducted two-sample t-tests for each task, and all differences between "Ours" and "w. latency" are statistically significant (p < 0.001).
>
>
> [1] Si, Zilin, et al. "Tilde: Teleoperation for dexterous in-hand manipulation learning with a deltahand." Robotics: Science and Systems (RSS 2024).
>
> [2] Bai, Fengshuo, et al. "Retrieval dexterity: Efficient object retrieval in clutters with dexterous hand." arXiv preprint arXiv:2502.18423 (2025).
>
> [3] Qi, Haozhi, et al. "In-hand object rotation via rapid motor adaptation." Conference on Robot Learning (CoRL 2023).
>
> [4] Suresh, Sudharshan, et al. "NeuralFeels with neural fields: Visuotactile perception for in-hand manipulation." Science Robotics (2024)
>
> [5] Zhang, Jialiang, et al. "Dexgraspnet 2.0: Learning generative dexterous grasping in large-scale synthetic cluttered scenes." 8th Annual Conference on Robot Learning. (CoRL 2024).

---

> > ### Author Response · Authors · 2025-08-07
> > **Please let us know whether whether we address all the questions**
> >
> > Dear reviewer,
> >
> > We appreciate your reviews and comments. We hope our responses address your questions.
> >
> > As the deadline is only one day away, please let us know if you have further questions after reading our rebuttal.
> >
> > We hope to address all the potential issues during the discussion period.
> >
> > Thank you

---

### Official Review · Reviewer_S1H9 · 2025-07-01

**Clarity:** 3
**Significance:** 2
**Originality:** 2
**Rating:** 4
**Confidence:** 3

**Summary:**

This paper presents RAPID Hand, a 20-DoF robotic hand platform that is open-source and low-cost. It integrates vision, tactile, and proprioceptive sensors, and uses a teleoperation interface for data collection.

 Main Contributions:

Designs a 20-DoF hand with parallel MCP joints and human-like motion.

Integrates and synchronizes vision, tactile, and proprioceptive sensors.

Provides a teleoperation interface for demonstration data collection.

Makes the hardware and software open-source and affordable.

**Questions:**

1. The tactile sensors are limited to the fingertips. Can the authors discuss how this impacts tasks involving palm contact?
2. The teleoperation interface not considered force feedback. Have the authors evaluated how this limitation affects quality?

**Ethical Concerns:**

["NO or VERY MINOR ethics concerns only"]

**Final Justification:**

addressed my concerns, I keep my initial rating

**Limitations:**

please refer to the Weaknesses and questions

**Quality:**

3

**Strengths And Weaknesses:**

Strengths：

Open-source, low-cost, and easy to replicate for the research community.

Hardware-level integration and synchronization of vision, tactile, and proprioceptive sensing; ensures high data fidelity.

Weaknesses：

Evaluation is mainly on a small set of tasks; broader real-world generalization needs further study.

Lacks force feedback for teleoperation, reducing realism of remote control.

---

> ### Author Rebuttal · Authors · 2025-07-30
>
> We thank the reviewer for the detailed reading of our paper and constructive suggestions! We hope our responses adequately address the following questions raised about our work. Please let us know if there’s anything we can clarify further.
>
> ```
> 1. Evaluation is mainly on a small set of tasks; broader real-world generalization needs further study.
> ```
> **Reply**: For autonomous manipulation with imitation learning, most existing SOTA work [1, 2, 3] focuses on pick-and-place, while our tasks, including in-hand rolling, translation, and multi-finger retrieval, involve more complex, contact-rich manipulation. We select them as atomic skills to test stability, sensing consistency, and dexterous control, providing a practical benchmark for evaluating RAPID Hand’s 20-DoF design and synchronized multimodal sensing.
>
> For dexterous teleoperation, a concurrent SOTA work on tool use [4] relaxes several conditions, such as placing tools in grasp friendly poses, assuming the tool is already in hand, or fixing the arm for stability. Our preliminary experiments under similar settings suggest that RAPID Hand can also support such capabilities.
>
> More complex tasks, such as full tool use or fine assembly, would further extend the platform’s potential. Progress toward these tasks remains constrained by the difficulty of collecting safe, high-quality demonstrations, particularly without tactile feedback [5]. As a natural next step, we are exploring **RAPID Glove**, a high-**R**esolution, **A**ffordable, ha**P**tic **I**nterface for **D**emonstrating contact-rich dexterous manipulation. While this is beyond the scope of the present paper, we see it as an important direction for extending the platform’s capabilities.
>
>
>
> ```
> 2. The teleoperation interface lacks force feedback, reducing realism of remote control; have the authors evaluated how this limitation affects quality?
> ```
> **Reply**: Tactile feedback can improve the realism and quality of teleoperation. In this work, we use a Vision Pro–based interface to ensure accessibility and ease of use. While it does not provide force feedback, it allows us to collect stable and high-quality demonstrations for all evaluated tasks.
>
> We do not directly quantify the impact of missing force feedback, as such an evaluation would require high-resolution haptic gloves, which are currently prohibitively expensive (USD 15,000–75,000) and bulky (e.g., HaptX [6], Meta [7]). Instead, we evaluate its effect through repeated teleoperation trials and qualitative monitoring of task outcomes. Based on these observations, the absence of tactile feedback can occasionally result in excessive force, such as surface damage during rolling.
>
>
>
>
> ```
> 3. The tactile sensors are limited to the fingertips. Can the authors discuss how this impacts tasks involving palm contact?
> ```
> **Reply**:
> In this work, we limit tactile sensing to the fingertips due to cost and integration constraints. Full-palm coverage requires custom-shaped or curved sensors, which remain costly (e.g., ReSkin [8], Paxini [9], Xela Robotics [10], Tencent [11]). Prior work shows that palm sensing can benefit certain tasks; for example, ReSkin applied to the Allegro Hand’s palm achieved nearly 100% success in in-hand translation [12].
>
> In our case, the evaluated tasks: rolling, translation, and multi-finger retrieval, have so far remained reliable without palm sensors. For in-hand translation, performance largely results from RAPID Hand’s natural lateral finger motion and fingertip tactile feedback, complemented by wrist-mounted vision. From a cost-effectiveness perspective, we currently refrain from deploying palm sensors, as they require complex custom designs that are expensive to implement.
>
> [1] Yang, Shiqi, et al. "Ace: A cross-platform visual-exoskeletons system for low-cost dexterous teleoperation, 2024." Conference on Robot Learning (CoRL 2024).
>
> [2] Cheng, Xuxin, et al. "Open-television: Teleoperation with immersive active visual feedback." Conference on Robot Learning (CoRL 2024).
>
> [3] Lin, Toru, et al. "Learning visuotactile skills with two multifingered hands." 2025 IEEE International Conference on Robotics and Automation (ICRA 2025).
>
> [4] Zorin, Anya, et al. "Ruka: Rethinking the design of humanoid hands with learning." Robotics: Science and Systems (RSS 2025).
>
>
> [5] Yin, Zhao-Heng, et al. "DexterityGen: Foundation controller for unprecedented dexterity." Robotics: Science and Systems (RSS 2025).
>
> [6] https://g1.haptx.com/.
>
> [7] https://research.facebook.com/publications/linking-haptic-parameters-to-the-emotional-space-for-mediated-social-touch/
>
>
> [8] https://ai.meta.com/blog/reskin-a-versatile-replaceable-low-cost-skin-for-ai-research-on-tactile-perception/
>
> [9] https://paxini.com/
>
> [10] https://www.xelarobotics.com/tactile-sensors
>
> [11] Lu, Peng, et al. "Thermoformed electronic skins for conformal tactile sensor arrays." 2024 IEEE International Conference on Robotics and Automation (ICRA 2024).
>
> [12] Yin, Jessica, et al. "Learning in-hand translation using tactile skin with shear and normal force sensing." 2025 IEEE International Conference on Robotics and Automation (ICRA 2025).

---

> > ### Author Response · Authors · 2025-08-07
> > **Please let us know if you have additional questions after reading our response**
> >
> > We appreciate your reviews and comments. We hope our responses address your concerns. Please let us know if you have further questions after reading our rebuttal.
> >
> > We hope to address all the potential issues during the discussion period.
> >
> > Thank you

---

> > ### Comment · Area_Chair_SBKr · 2025-08-08
> > **Please respond to author comments**
> >
> > Hi, Please respond to the author comments. They have provided a detail update and it'd be good to provide them constructive feedback.

---

### Decision · Program_Chairs · 2025-09-17

**Decision:**

Accept (poster)

**Comment:**

This paper introduces the RAPID Hand, a low-cost, 20-DoF robotic hand with integrated multimodal perception (vision/tactile) for dexterous manipulation. The hand design has parallel MCP joints and natural human-like kinematics and seems superior to alternatives like the LEAP or Allegro hand. The authors also provide a carefully designed hardware integration and software stack for sensing and teleoperation. They then validate this via policy learning on a number of high dexterity problems. Given the timeliness of the field of dexterous manipulation and the urgent need for an easily usable, dexterous hardware platform - this paper fills an important need for the field of dexterous manipulation. The reviewers did bring up some good points about haptic feedback, more clearly highlighting the advantages of the RAPID hand over others, lack of a clear tutorial for asssembly and limited task validation. These concerns should be addressed in a final version of the submission.